# Alignment-Guided Score Matching
# for Text-to-Image Alignment in Diffusion Models

**Jaa-Yeon Lee** [* 1]   **Yeobin Hong** [* 1]   **Taesung Kwon** [1]   **Jong Chul Ye** [1]

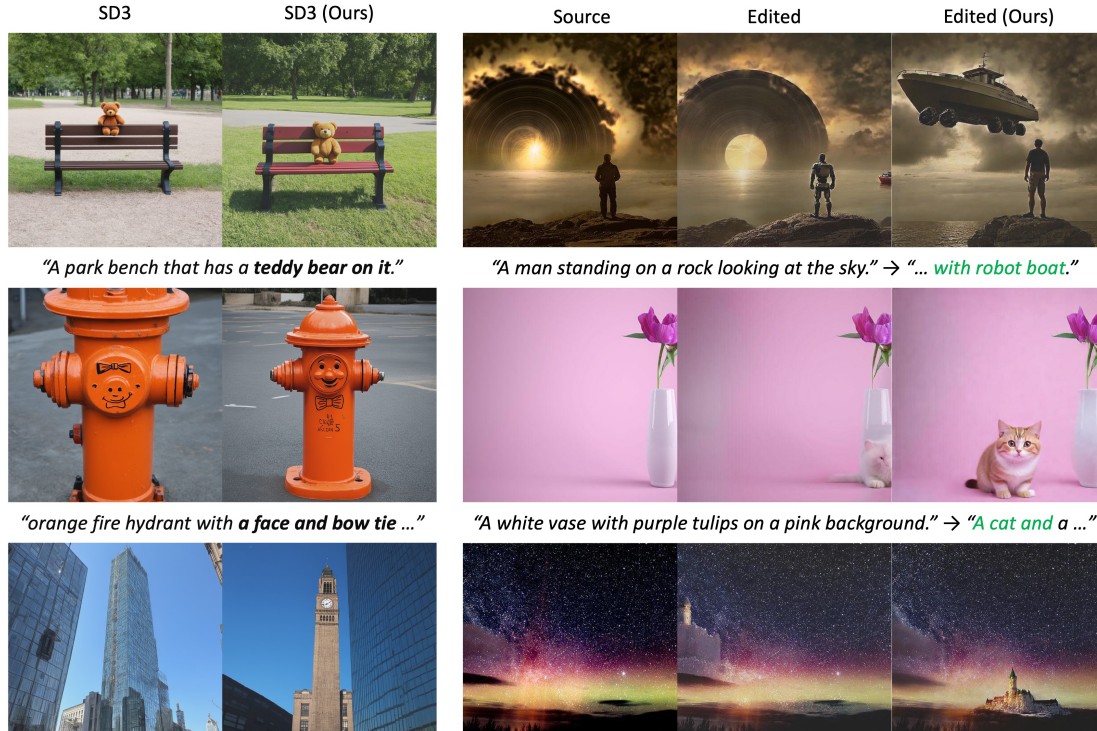

SD3            SD3 (Ours)                   Source          Edited          Edited (Ours)

*"A park bench that has a **teddy bear on it**."*          *"A man standing on a rock looking at the sky." → "… with robot boat."*

*"orange fire hydrant with **a face and bow tie** …"*          *"A white vase with purple tulips on a pink background." → "A cat and a …"*

*"… tall **clock tower** next to a tall glass building."*          *"The arctic sky." → "A castle in the arctic sky."*

*Figure 1.* Representative results for text-to-image generation and image editing. Our alignment-guided fine-tuning improves semantic consistency between text and image by training soft tokens within the score-matching framework.

## Abstract

Diffusion models generate highly realistic images but often struggle with precise text–image alignment. While recent post-training methods improve alignment using external rewards or human preference signals, their performance heavily depends on reward quality and does not directly address alignment within the diffusion process itself. Recent reward-free approaches such as SoftREPA demonstrate that optimizing soft text tokens via

contrastive learning can effectively improve text-image representation alignment, outperforming standard parameter-efficient fine-tuning baselines. However, the contrastive formulation can excessively penalize negative pairs, which manifests as characteristic failure cases such as overcounting and repetition. To address this issue, we propose a lightweight, reward-free post-training method that refines soft tokens by integrating contrastive alignment guidance directly into the score-matching objective of diffusion models. By assigning alignment directions at the score level, our approach mitigates these limitations and yields more coherent and semantically faithful generations. Experiments show that our method matches SoftREPA while substantially improving

*Equal contribution   [1]Graduate School of AI, KAIST, South Korea. Correspondence to: Jong Chul Ye <jong.ye@kaist.ac.kr>.

*Proceedings of the 43rd International Conference on Machine Learning*, Seoul, South Korea. PMLR 306, 2026. Copyright 2026 by the author(s).

its failure cases, achieving over 35% improvement in counting accuracy on the GenEval benchmark. Our method is seamlessly applicable to existing diffusion backbones (SD1.5, SDXL, and SD3), and is complementary to existing RL-based diffusion post-training methods. Project page: https://jaayeon.github.io/AGSM/

# 1. Introduction

Diffusion models have achieved remarkable progress in high-fidelity image generation (Peebles & Xie, 2023; Esser et al., 2024; Podell et al., 2023). To further align their generative behavior with desired outcomes, recent work has explored post-training techniques such as policy gradient and preference optimization (Black et al., 2023; Xu et al., 2023; Clark et al., 2023; Fan et al., 2023; Wallace et al., 2024). However, most existing approaches rely on human preference annotations (Wallace et al., 2024; Karthik et al., 2024; Zhu et al., 2025; Liang et al., 2025) or externally designed reward models (Fan et al., 2023; Black et al., 2023; Xu et al., 2023; Clark et al., 2023). As a result, their effectiveness depends critically on reward quality and data availability, leaving the intrinsic text–image alignment signals within diffusion models underexplored.

Recent studies have begun to revisit text–image alignment by leveraging diffusion models' internal representations and score-matching dynamics (Xian et al., 2025; Lee et al., 2025a). In particular, SoftREPA (Lee et al., 2025a) demonstrated the potential of optimizing lightweight soft text tokens to maximize the mutual information between modalities by leveraging the diffusion score-matching loss as a proxy for alignment. However, we identify a fundamental instability in this contrastive formulation: while it minimizes score-matching loss for positive pairs, it simultaneously maximizes it for negative pairs. This adversarial pushing often forces the soft tokens to represent off-manifold regions, manifesting in characteristic failure cases such as object repetition, over-counting, and semantic incoherence.

In parallel, recent advances in diffusion-based preference optimization provide a promising direction. Diffusion-DPO (Direct Preference Optimization) (Wallace et al., 2024) formulates preference alignment within the diffusion objective using the Bradley–Terry model (Bradley & Terry, 1952). DSPO (Direct Score Preference Optimization) (Zhu et al., 2025) further integrates preference learning into the score-matching framework. These approaches indicate that modeling preferences at the score level enables stable alignment while preserving the underlying diffusion dynamics.

Building on these insights, we propose **Alignment-Guided Score Matching**, a reward-free post-training framework optimizing soft tokens that addresses contrastive instability by explicitly guiding both positive and negative text–image pairs within the score-matching objective. We formulate text–image alignment as preference learning under a Plackett–Luce (PL) model (Luce et al., 1959), where alignment preferences are derived from the diffusion model's intrinsic log-likelihood without external rewards. Unlike prior approaches that penalize negative pairs implicitly, our method assigns explicit score-level guidance using separate soft tokens $(\psi^+, \psi^-)$ for positive and negative semantic regions, preventing off-manifold drift and preserving generative fidelity.

Our main contributions are as follows:

- **Reward-free Plackett-Luce Formulation**: We formulate text-image alignment as reward fine-tuning over diffusion scores. Employing a PL model, we enable a reward-free post-training objective that leverages the model's internal priors.

- **Stability via Explicit Negative Guidance**: We mitigate the unbounded divergence of prior contrastive loss by assigning explicit, bounded preference directions to negative samples, preventing the failure cases of SoftREPA.

- **Efficiency and Versatility**: The proposed approach is lightweight, model-agnostic, and complementary to existing RL-based diffusion post-training methods.

# 2. Preliminaries

**SoftREPA** SoftREPA (Lee et al., 2025a) fine-tunes diffusion models by aligning text and image representations through contrastive learning. Given a soft text token $s$ and a paired sample $(x, c)$, the similarity score is defined as

$$\tilde{\ell}(\boldsymbol{x}, \boldsymbol{c}, \boldsymbol{s}) = \exp\left(-\mathbb{E}_{t,\boldsymbol{\epsilon}}\left[\frac{\|\epsilon_\theta(\boldsymbol{x}_t, t, \boldsymbol{c}, \boldsymbol{s}) - \boldsymbol{\epsilon}_t\|_2^2}{\tau(t)}\right]\right), \quad (1)$$

where $\epsilon_\theta$ is the noise prediction of the diffusion model, $\tau(t)$ denotes a temperature-scaled time weighting, and $\boldsymbol{\epsilon}_t$ is the Gaussian noise at step $t$. In practice, SoftREPA approximates the expectation with a Monte Carlo estimate using sampled $t$ and $\boldsymbol{\epsilon}$ during training. The soft-token training objective adopts a contrastive form:

$$\mathcal{L}(\boldsymbol{s}) = -\mathbb{E}_{\substack{(\boldsymbol{x},\boldsymbol{c})\sim p_{\text{data}},\\ t\sim U(0,1),\\ \boldsymbol{\epsilon}\sim\mathcal{N}(\boldsymbol{0},\mathbf{I})}} \log \frac{\exp\left(\tilde{\ell}(\boldsymbol{x}, \boldsymbol{c}, \boldsymbol{s})\right)}{\sum_j \exp\left(\tilde{\ell}(\boldsymbol{x}, \boldsymbol{c}^j, \boldsymbol{s})\right)}, \quad (2)$$

where $\boldsymbol{c}^j$ denotes negative text pairs within the minibatch. This objective optimizes the soft token $s$ to maximize text–image representation alignment by increasing the mutual information between the two modalities.

**DSPO** Direct Score Preference Optimization (DSPO) (Zhu et al., 2025) fine-tunes diffusion models by directly incorporating human preference signals into the score-matching framework. Given a text-conditioned diffusion model $p_\theta(\boldsymbol{x}_t|\boldsymbol{c})$ and a preference pair $(\boldsymbol{x}_t, \boldsymbol{x}_t^l, \boldsymbol{c})$, human preference is modeled by the Bradley–Terry formulation (Bradley & Terry, 1952):

$$p(\boldsymbol{y}|\boldsymbol{x}_t, \boldsymbol{c}) = \sigma\left(r(\boldsymbol{x}_t, \boldsymbol{c}) - r(\boldsymbol{x}_t^l, \boldsymbol{c})\right), \qquad (3)$$

where $p(\boldsymbol{y}|\boldsymbol{x}_t, \boldsymbol{c})$ denotes the probability that $(\boldsymbol{x}_t, \boldsymbol{c})$ is preferred to $(\boldsymbol{x}_t^l, \boldsymbol{c})$ and $r(\boldsymbol{x}_t, \boldsymbol{c})$ is an implicit reward estimated from DiffusionDPO (Wallace et al., 2024). The DSPO objective aligns the diffusion score with the human-preferred score, $\nabla_{\boldsymbol{x}_t} \log p(\boldsymbol{x}_t|\boldsymbol{c}, \boldsymbol{y})$ using Bayes' rule, as

$$\min_\theta \|\nabla \log p_\theta(\boldsymbol{x}_t|\boldsymbol{c}) - (\nabla \log p(\boldsymbol{x}_t|\boldsymbol{c}) + \gamma \nabla \log p(\boldsymbol{y}|\boldsymbol{x}_t, \boldsymbol{c})\|_2^2 \tag{4}$$

where $\gamma$ controls the preference strength. The implicit reward can be expressed as a log-density ratio between the current and reference models:

$$r(\boldsymbol{x}_t, \boldsymbol{c}) = \lambda_t \log \frac{p_\theta(\boldsymbol{x}_{t-1}|\boldsymbol{x}_t, \boldsymbol{c})}{p_{\text{ref}}(\boldsymbol{x}_{t-1}|\boldsymbol{x}_t, \boldsymbol{c})}. \tag{5}$$

Combining Eq. (3) and Eq. (5) into Eq. (4), DSPO training objective becomes:

$$\min_\theta \left\| \boldsymbol{\epsilon}_{\theta,t} - \boldsymbol{\epsilon}_t - \lambda_t\, w(\boldsymbol{x}_t, \boldsymbol{x}_t^l, \boldsymbol{c})(\boldsymbol{\epsilon}_{\theta,t} - \boldsymbol{\epsilon}_{\text{ref},t}) \right\|_2^2, \tag{6}$$

where $w(\boldsymbol{x}_t, \boldsymbol{x}_t^l, \boldsymbol{c}) = 1 - \sigma\left(r(\boldsymbol{x}_t, \boldsymbol{c}) - r(\boldsymbol{x}_t^l, \boldsymbol{c})\right)$ is a preference score-based weighting term that modulates the guidance induced by the discrepancy between the online and reference scores.

# 3. Alignment-Guided Score Matching

Since the similarity score in Eq. (1) is a strictly decreasing function, minimizing Eq. (2) may lead to increasing the score matching loss $\|\boldsymbol{\epsilon}_\theta(\boldsymbol{x}_t, t, \boldsymbol{c}^j, \boldsymbol{s}) - \boldsymbol{\epsilon}_t\|_2^2$ for negative pairs $\boldsymbol{c}^j$. Therefore, the SoftREPA objective does not constrain negative pairs to remain on the diffusion manifold, allowing unbounded divergence of the denoising error.

To circumvent the instabilities of contrastive pushing, we propose a guidance-based framework that treats text alignment as a bounded preference optimization problem. Our approach consists of three components: (i) a normalized alignment reward derived via a Plackett-Luce formulation (Section 3.1), (ii) a modified score-matching objective that transforms the target score for both positive and negative pairs (Section 3.2), and (iii) a dual-token training scheme that explicitly separates positive and negative guidance with stability analysis (Section 3.3).

## 3.1. Alignment Reward via Plackett–Luce Modeling

To define the probability that an image $\boldsymbol{x}_t$ is aligned with a specific text $\boldsymbol{c}$ among a set of candidates $\{\boldsymbol{c}^i\}$, we employ the Plackett-Luce (PL) model (Luce et al., 1959). This formulation generalizes the pairwise Bradley-Terry model used in Eq. (3) to a multi-class preference framework:

$$p(z=1|\boldsymbol{x}_t, \boldsymbol{c}) = \frac{\exp(r(\boldsymbol{x}_t, \boldsymbol{c}))}{\sum_i \exp(r(\boldsymbol{x}_t, \boldsymbol{c}^i))}, \tag{7}$$

where $z$ serves as a binary random variable: $z=1$ indicates the pair $(\boldsymbol{x}_t, \boldsymbol{c})$ is aligned and $z=0$ indicates opposite.

Inspired by SoftREPA (Lee et al., 2025a), we define an implicit alignment reward as the expected conditional log-likelihood of the model reverse transition under DDPM (Ho et al., 2020) posterior:

$$r(\boldsymbol{x}_t, \boldsymbol{c}) := \lambda_t\, \mathbb{E}_{q(\boldsymbol{x}_{t-1}|\boldsymbol{x}_t, \boldsymbol{x}_0)}[\log p_\theta(\boldsymbol{x}_{t-1} \mid \boldsymbol{x}_t, \boldsymbol{c})], \tag{8}$$

where $\lambda_t$ controls the reward scale. Thus, higher reward is assigned to text–image pairs whose reverse transition is better predicted under condition $\boldsymbol{c}$, without requiring an external reward model.

## 3.2. Alignment-Guided Score Matching

**Target Distribution.** To maximize the alignment of the joint pdf of $\boldsymbol{x}_t$ and $\boldsymbol{c}$, we divide data into positive and negative subsets $(\mathcal{D}^+, \mathcal{D}^-)$ using a binary random variable $z$. $\mathcal{D}^+$ consists of aligned text–image pairs $(z=1)$, whereas $\mathcal{D}^-$ consists of mismatched pairs $(z=0)$. The goal is to increase the probability in aligned regions while suppressing it in the mismatched regions through explicit score modification.

The corresponding tilted target conditional distributions are defined as

$$p_t^+(\boldsymbol{x}_t|\boldsymbol{c}) := p_t(\boldsymbol{x}_t|\boldsymbol{c}, z=1) \propto p_t(\boldsymbol{x}_t|\boldsymbol{c})\, p(z=1|\boldsymbol{x}_t, \boldsymbol{c})^{\gamma^+},$$
$$p_t^-(\boldsymbol{x}_t|\boldsymbol{c}) := p_t(\boldsymbol{x}_t|\boldsymbol{c}, z=0) \propto p_t(\boldsymbol{x}_t|\boldsymbol{c})\, p(z=1|\boldsymbol{x}_t, \boldsymbol{c})^{-\gamma^-} \tag{9}$$

where $\gamma^+$ and $\gamma^-$ regulate the influence of alignment reward on the resulting posterior. This formulation is closely related to classifier-free guidance (CFG) (Ho & Salimans, 2022), which tilts the sampling distribution with weighted posterior probability, $p_\theta(\boldsymbol{x}|\boldsymbol{c})p_\theta(\boldsymbol{c}|\boldsymbol{x}_t)^w$. Similarly, the negative branch acts as a repulsive tilting term analogous to negative prompting (Gandikota et al., 2023). The resulting inverse weighting serves as a direction-preserving surrogate for the Bayes-consistent negative guidance term $p(z=0|\boldsymbol{x}_t, \boldsymbol{c})$, differing only by a positive scaling factor (Appendix A).

A unified expression of the modified target distribution is

$$\tilde{p}_t(\boldsymbol{x}_t|\boldsymbol{c}, z) := \mathbf{1}\{z=1\}p_t^+(\boldsymbol{x}_t|\boldsymbol{c}) + \mathbf{1}\{z=0\}p_t^-(\boldsymbol{x}_t|\boldsymbol{c}). \tag{10}$$

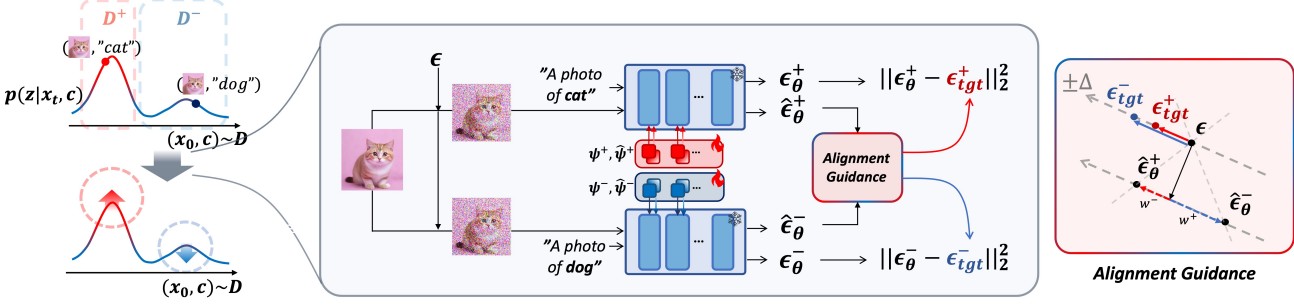

*Figure 2.* Alignment-Guided Score Matching improves text–image alignment by increasing alignment rewards for positive pairs and decreasing those for negative pairs. Noise predictions $\epsilon_\theta^+$ and $\epsilon_\theta^-$ are conditioned on positive and negative soft tokens $(\psi^+, \psi^-)$. Target noise is adjusted using alignment guidance derived from implicit-reward–weighted EMA predictions $(\hat{\epsilon}_\theta^+, \hat{\epsilon}_\theta^-)$. For clarity, the figure shows a single negative pair, while multiple negatives are used during training.

Taking the gradient with respect to $\boldsymbol{x}_t$ gives the new target score:

$$\nabla \log \tilde{p}_t(\boldsymbol{x}_t|\boldsymbol{c},z) = \nabla \log p_t(\boldsymbol{x}_t|\boldsymbol{c}) + \gamma_z \nabla \log p(z{=}1|\boldsymbol{x}_t,\boldsymbol{c}), \quad (11)$$

where $\gamma_z = \gamma^+ \mathbf{1}\{z{=}1\} - \gamma^- \mathbf{1}\{z{=}0\}$. The first term corresponds to the standard diffusion score, while the second term explicitly pushes samples in the direction of higher reward gradients when it comes to positive pairs, pushes samples in the opposite direction when it comes to negative pairs, encouraging $\boldsymbol{x}_t$ to move toward better-aligned image–text regions.

**Training Objective.** We train learnable soft tokens $\psi_z$ to match the new target score:

$$\min_{\psi_z} \mathbb{E}\Big[w(t)\|\nabla \log p_{t,\theta}^{\psi_z}(\boldsymbol{x}_t|\boldsymbol{c}) - \nabla \log \tilde{p}_t(\boldsymbol{x}_t|\boldsymbol{c},z)\|_2^2\Big], \quad (12)$$

where $w(t)$ is a time-dependent weighting function. We denote by $p_{t,\theta}^{\psi_z}(\boldsymbol{x}_t|\boldsymbol{c})$ the diffusion model with frozen backbone $\theta$ conditioned on soft token $\psi_z$.

To compute the gradient term in Eq. (11), we differentiate the PL likelihood in Eq. (7) with respect to $\boldsymbol{x}_t$:

$$\nabla \log p(z{=}1|\boldsymbol{x}_t,\boldsymbol{c}) = \nabla r(\boldsymbol{x}_t,\boldsymbol{c}) - \sum_i w_i \nabla r(\boldsymbol{x}_t,\boldsymbol{c}^i), \quad (13)$$

where $w_i = \frac{\exp(r(\boldsymbol{x}_t,\boldsymbol{c}^i))}{\sum_j \exp(r(\boldsymbol{x}_t,\boldsymbol{c}^j))}$ represents the normalized reward weight over textual alternatives. This equation shows that the gradient of the PL likelihood compares the reward gradient of the current pair $(\boldsymbol{x}_t,\boldsymbol{c})$ against the weighted average of competing candidates, producing a contrast signal for alignment.

Since diffusion models parameterize the reverse denoising process using a neural network as in DDPM (Ho et al.,

**Algorithm 1** Alignment-Guided Score Matching

**Require:** Dataset $\mathcal{D}$; backbone $\theta$; soft tokens $\psi^\pm$; EMA soft tokens $\hat{\psi}^\pm$; guidance scales $\gamma^\pm$
1: **while** not converged **do**
2:     ▷ Sample positive and negative pairs
3:     $(\boldsymbol{x}_0^i, \boldsymbol{c}^j)_{i,j=1}^B \sim \mathcal{D}$ {$\mathcal{D}^+$ if $i=j$, $\mathcal{D}^-$ if $i\neq j$}
4:     $t \sim U(0,1), \epsilon \sim \mathcal{N}(0,\mathbf{I})$
5:     ▷ use same $t$ and $\epsilon$ for all pairs
6:     $\boldsymbol{x}_t^i = \sqrt{\bar{\alpha}_t}\boldsymbol{x}_0^i + \sqrt{1-\bar{\alpha}_t}\epsilon, \forall i$
7:     $(\epsilon_{\text{pred}}^{(i,j)}, \epsilon_{\text{ema}}^{(i,j)}) \leftarrow \begin{cases} (\epsilon_{t,\theta}^{\psi^+}(\cdot), \epsilon_{t,\theta}^{\hat{\psi}^+}(\cdot)), & i=j \\ (\epsilon_{t,\theta}^{\psi^-}(\cdot), \epsilon_{t,\theta}^{\hat{\psi}^-}(\cdot)), & i \neq j \end{cases}$
            where $(\cdot) = (\boldsymbol{x}_t^i, \boldsymbol{c}^j)$
8:     $w_{i,j} = \text{Softmax}_j\big(-\|\epsilon_{\text{ema}}^{(i,j)} - \epsilon\|_2^2\big)$ {Eq.14, Eq.7}
9:     $\Delta_{\hat{\psi}}^{(i,j)} = \epsilon_{\text{ema}}^{(i,j)} - \sum_k w_{i,k}\epsilon_{\text{ema}}^{(i,k)}$ {Eq.13}
10:    $\epsilon_{\text{tgt}}^{(i,j)} \leftarrow \epsilon + \begin{cases} +\gamma^+ \tilde{A}(t)\Delta_{\hat{\psi}}^{(i,j)}, & i=j \\ -\gamma^- \tilde{A}(t)\Delta_{\hat{\psi}}^{(i,j)}, & i \neq j \end{cases}$
11:    ▷ Update $\psi^\pm$ and $\hat{\psi}^\pm$
12:    $\mathcal{L}(\psi^\pm) \propto \sum_{i,j}\|\epsilon_{\text{pred}}^{(i,j)} - \epsilon_{\text{tgt}}^{(i,j)}\|_2^2$
13: **end while**

2020), we instantiate the alignment reward using denoising error:

$$r(\boldsymbol{x}_t,\boldsymbol{c}) = -\frac{A(t)}{2}\|\epsilon_\theta^{\hat{\psi}_z}(\boldsymbol{x}_t,t,\boldsymbol{c}) - \epsilon\|_2^2, \quad (14)$$

where $\alpha_t{=}1-\beta_t$, $\bar{\alpha}_t{=}\prod_{s=1}^t \alpha_s$, and $A(t) = \frac{\lambda_t \beta_t}{\alpha_t(1-\bar{\alpha}_{t-1})}$ (see Appendix B for details). We compute the reward using EMA-updated soft tokens($\hat{\psi}_z$) to stabilize the alignment signal. A lower denoising error corresponds to a higher reward, directly coupling text–image alignment with diffusion consistency.

By substituting Eq. (11), Eq. (14), and Eq. (13) into the score-matching objective in Eq. (12), we obtain the final Alignment-Guided Score Matching loss:

$$\min_{\psi_z} \mathbb{E}\Big[\|\epsilon_{t,\theta}^{\psi_z} - (\epsilon_t + \gamma_z \tilde{A}(t)(\epsilon_{t,\theta}^{\hat{\psi}_z} - \sum_i w_i \epsilon_{t,\theta}^{\hat{\psi}_z,i}))\|^2\Big]. \quad (15)$$

Here, $\epsilon_{t,\theta}^{\hat{\psi}_z,i}$ denotes the denoising prediction conditioned on the $i$-th text candidate $c^i$. We omit the timestep weighting for brevity, with $\tilde{A}(t) = \frac{\lambda_t \beta_t \sqrt{1-\bar{\alpha}_t}}{\alpha_t(1-\bar{\alpha}_{t-1})}$. A comprehensive derivation is provided in Appendix C.

For the flow model, the Alignment-Guided Score Matching loss can be formulated as

$$\min_{\psi_z} \mathbb{E}\Big[\big\| v_{t,\theta}^{\psi_z} - \big(v_t + \gamma_z B(t)\big(v_{t,\theta}^{\hat{\psi}_z} - \sum_i w_i\, v_{t,\theta}^{\hat{\psi}_z,i}\big)\big\|_2^2\Big], \tag{16}$$

which closely resembles the alignment-guided loss for the diffusion model (Eq. (15)). A detailed derivation is given in Appendix D.

### 3.3. Negative Sample Optimization and its Stability

**Dual-Token Parameterization.** To effectively capture both generative capability and alignment performance, we decouple the positive and negative alignment guidance through separate soft token optimization. Separate soft tokens ($\psi^+$ and $\psi^-$) are updated for the corresponding positive and negative data pairs (Figure 2). Concretely, the resulting objective Eq. (15) can be rewritten as:

$$\min_{\psi^+,\psi^-} \Big( \mathbb{E}_{(\boldsymbol{x},\boldsymbol{c})\sim\mathcal{D}^+}\Big[\big\|\epsilon_{t,\theta}^{\psi^+} - \big(\epsilon_t + \gamma^+ \tilde{A}(t)\,\Delta_{\hat{\psi}_z}\big)\big\|_2^2\Big]$$
$$+ \mathbb{E}_{(\boldsymbol{x},\boldsymbol{c})\sim\mathcal{D}^-}\Big[\big\|\epsilon_{t,\theta}^{\psi^-} - \big(\epsilon_t - \gamma^- \tilde{A}(t)\,\Delta_{\hat{\psi}_z}\big)\big\|_2^2\Big] \Big), \tag{17}$$

where $\Delta_{\hat{\psi}_z} = \epsilon_{t,\theta}^{\hat{\psi}_z} - \sum_i w_i\,\epsilon_{t,\theta}^{\hat{\psi}_z,i}$. Decoupling the parameters provides a mechanism to partition the alignment and contrastive signals, allowing for targeted semantic refinement without the risk of over-optimizing the negative pairs at the expense of generative fidelity. The complete training algorithm is summarized in Algorithm 1.

**Stability of Alignment Guidance.** Unlike SoftREPA, whose log-sum-exp contrastive objective admits descent directions that inflate negative denoising errors, our alignment-guided objective introduces a normalized preference correction within score matching. When the matched candidate dominates in Plackett–Luce (PL) form, $\Delta_{\hat{\psi}_z}$ becomes small so that the target score reduces toward the standard denoising objective. For negative pairs, $\Delta_{\hat{\psi}_z}$ contributes only through a normalized weighted correction term.

Concretely, the alignment term $\gamma_z \nabla_{\boldsymbol{x}_t} \log p(z|\boldsymbol{x}_t, \boldsymbol{c})$ takes the PL form $\nabla r(\boldsymbol{x}_t, \boldsymbol{c}) - \sum_i w_i \nabla r(\boldsymbol{x}_t, \boldsymbol{c}^i)$, whose norm is bounded by a weighted combination of reward gradients,

$$\big\|\nabla \log p(z|\boldsymbol{x}_t,\boldsymbol{c})\big\| \le \big\|\nabla r(\boldsymbol{x}_t,\boldsymbol{c})\big\| + \sum_i w_i \big\|\nabla r(\boldsymbol{x}_t,\boldsymbol{c}^i)\big\|. \tag{18}$$

With the reward instantiated as a scaled denoising error, the resulting correction remains finite and does not encourage unbounded growth of negative diffusion losses. In practice, this leads to substantially more stable training dynamics compared to SoftREPA, which exhibits gradual degradation without early-stopping (Training Dynamics Analysis in Section 4).

## 4. Experiments

**Implementation Details.** We conducted experiments on SD1.5, SDXL, and SD3. For SD1.5 and SDXL, we trained soft tokens applied to the Down and Middle blocks of the UNet backbone, with 8 (4 positive and 4 negative) soft text tokens. For SD3, we trained 8 soft text tokens on the upper 5 transformer layers, which is the same configuration as Soft-REPA. The batch size was set to 16, which makes 3 negative prompts for each text-image pair across all models. Regarding SoftREPA, we used official checkpoints trained with larger negative pools: 7 negatives per positive (batch 64) for SD1.5/SDXL, and 3 negatives (batch 16) for SD3. During sampling, we dropped the negative soft tokens ($\psi^-$) and used only the positive soft tokens ($\psi^+$) for both conditional and unconditional generation. The positive and negative guidance scales ($\gamma^+, \gamma^-$) were set to $(1, 1)$ for SD1.5 and SDXL, and $(1, 0.1)$ for SD3. For simplicity, we set the time-dependent reward scale $\lambda_t$ so that $\tilde{A}(t) = 1$ during training. Further implementation details are provided in Appendix G.

**Training Dynamics Analysis.** We further analyze training stability against SoftREPA by tracking validation ImageReward throughout post-training. As shown in Figure 4, SoftREPA reaches peak ImageReward at early iterations and then substantially degrades, whereas our method maintains stable performance over longer training. In particular, Soft-REPA's loss continues to decrease even when ImageReward drops, indicating over-optimization of the contrastive objective and the need for heuristic early stopping. By contrast, our PL-based score-matching objective uses a bounded and normalized correction, which reduces late-stage deterioration from amplified negative signals.

**Text to Image Generation.** We conducted text-to-image generation experiments comparing our method with the baseline and SoftREPA (Lee et al., 2025a) on SD1.5, SDXL, and SD3. All models were trained on the COCO-train dataset (Lin et al., 2014) and evaluated on the COCO-val and GenEval benchmarks (Ghosh et al., 2023).

In Table 1, our method achieves improved text–image alignment and image quality compared to the baselines. For the GenEval benchmark, we additionally compare against CaPO (Lee et al., 2025b) and RankDPO (Karthik et al., 2024), recent preference-based post-training approaches. Notably, our method significantly improves counting accuracy by $+35\%$, effectively mitigating the over-emphasizing

SD3      SoftREPA      Ours          SD3      SoftREPA      Ours

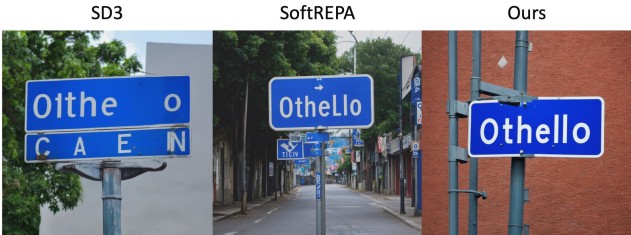
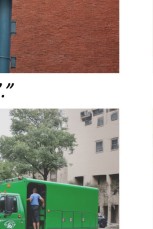
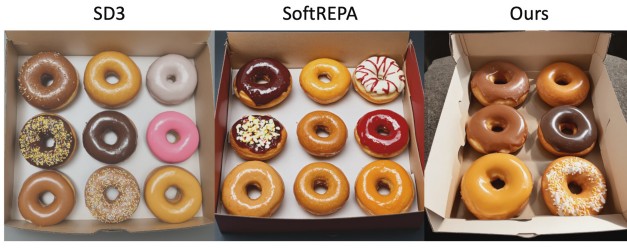

*"A blue and white street sign that reads **'Othello'**."*         *"A box contains **six donuts** with varying types of glazes and toppings."*

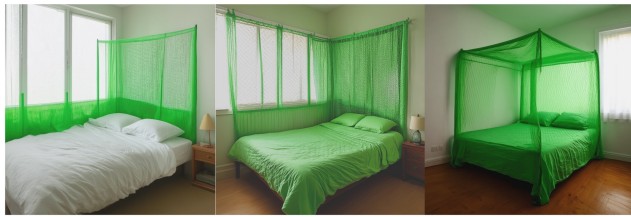
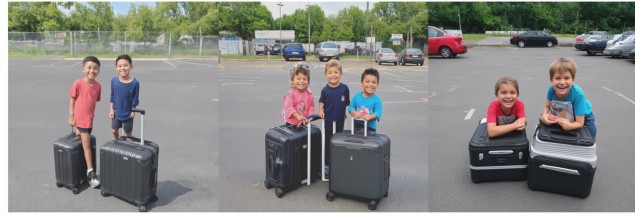

*"A green utility truck is parked on a street while **a man** climbs inside."*       *"**Two children** smiling on top of luggage in parking lot."*

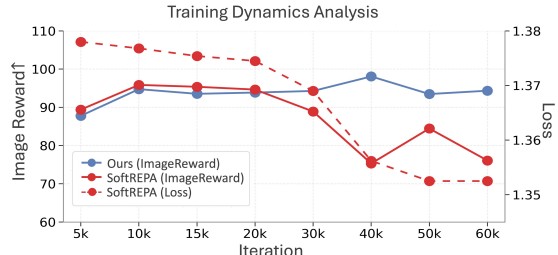
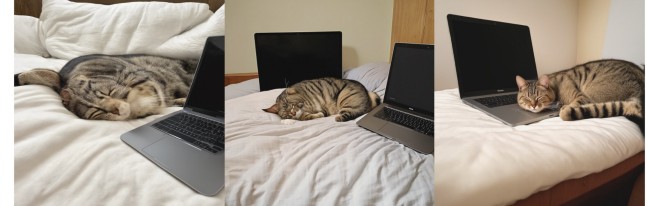
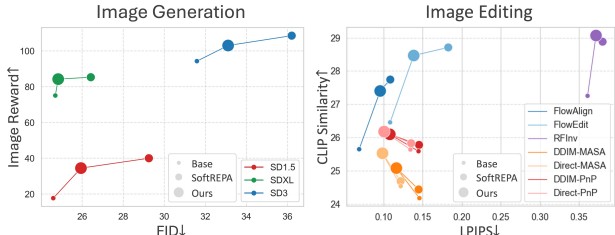

*"**A green netted bed** in a light filled bedroom."*          *"A cat sleeping on a bed next to **a laptop computer**."*

*Figure 3.* Qualitative comparison of text-to-image generation among SD3, SoftREPA, and our method. Prompts are sampled from the COCO validation set. Compared to SD3 and SoftREPA, our method produces images that better reflect the input text, while reducing common failure modes such as object repetition of SoftREPA.

*Figure 4.* Training stability comparison between AGSM (Ours) and SoftREPA. SoftREPA's validation ImageReward degrades despite decreasing training loss, while our method remains stable throughout the later stages of training.

*Figure 5.* Comparison between the baseline, SoftREPA, and our method on image generation (ImageReward vs. FID) and image editing (CLIP vs. LPIPS) tasks. Optimal performance corresponds to the top-left region of the plot.

behavior observed in prior methods.

In Figure 5-(left), we evaluate performance along two complementary axes: human preference metrics (ImageReward) and image quality and diversity (FID). While a trade-off exists between these metrics, our method consistently improves preference-aligned performance over the baseline while maintaining substantially better FID than SoftREPA. Detailed quantitative results are provided in Appendix E. Figure 3 presents qualitative comparisons among SD3, Soft-REPA (Lee et al., 2025a), and our method, showing that our approach reduces redundant object generation and adheres more faithfully to the given text conditions.

**Text Guided Image Editing.** We evaluate our method on PIE-Bench (Ju et al., 2023), a standard benchmark containing 700 images, containing source, target prompts and background mask. We compare our method against several training-free text-based editing baselines for SD1.5 and SD3. For SD1.5, we include PnP (Tumanyan et al., 2023) and MasaCtrl (Cao et al., 2023), evaluated with both direct (Ju et al., 2023) and DDIM (Song et al., 2021a) inversion. For SD3, we select methods representing different strategies: RF-Inversion (Rout et al., 2024) for an inversion-based approach and FlowEdit (Kulikov et al., 2024), FlowAlign (Kim et al., 2025) for inversion-free methods. Detailed experimental configurations are available in Appendix F.

| COCO val5K | | | | | |
|---|---|---|---|---|---|
| Model | ImageReward↑ | PickScore↑ | CLIP↑ | HPSv2↑ | FID↓ |
| SD1.5 | 17.72 | 21.47 | 26.4 | 25.08 | **24.59** |
| Ours | **34.50** | **21.59** | **27.23** | **25.66** | 25.94 |
| SDXL | 75.06 | 22.38 | 26.76 | 27.35 | **24.69** |
| Ours | **84.22** | **22.57** | **26.86** | **27.96** | 24.83 |
| SD3 | 94.27 | **22.54** | 26.30 | 28.09 | **31.59** |
| Ours | **103.3** | 22.39 | **27.00** | **28.22** | 34.08 |

| GenEval | | | | | | | |
|---|---|---|---|---|---|---|---|
| Model | # Trainable Params | Mean↑ | Single↑ | Two↑ | Counting↑ | Colors↑ | Position↑ | Color Attribution↑ |
| SD3 | - | 0.68 | 0.99 | 0.86 | 0.56 | 0.85 | 0.27 | 0.55 |
| CaPO (Lee et al., 2025b) | 2B | 0.71 | 0.99 | 0.87 | 0.63 | 0.86 | 0.31 | 0.59 |
| RankDPO (Karthik et al., 2024) | 2B | **0.74** | **1.00** | 0.90 | **0.72** | 0.87 | 0.31 | 0.66 |
| SoftREPA (Lee et al., 2025a) | 0.9M | 0.70 | **1.00** | 0.95 | 0.29 | **0.92** | 0.34 | 0.68 |
| Ours | 1.8M | 0.72 | **1.00** | 0.91 | 0.64 | 0.89 | 0.26 | 0.64 |

*Table 1.* Quantitative evaluation of T2I generation on SD1.5, SDXL, and SD3. Generation quality is evaluated on the COCO-val 5K (Lin et al., 2014) and GenEval (Ghosh et al., 2023) benchmark. ImageReward, CLIP, HPS, and LPIPS are scaled by $\times 10^2$.

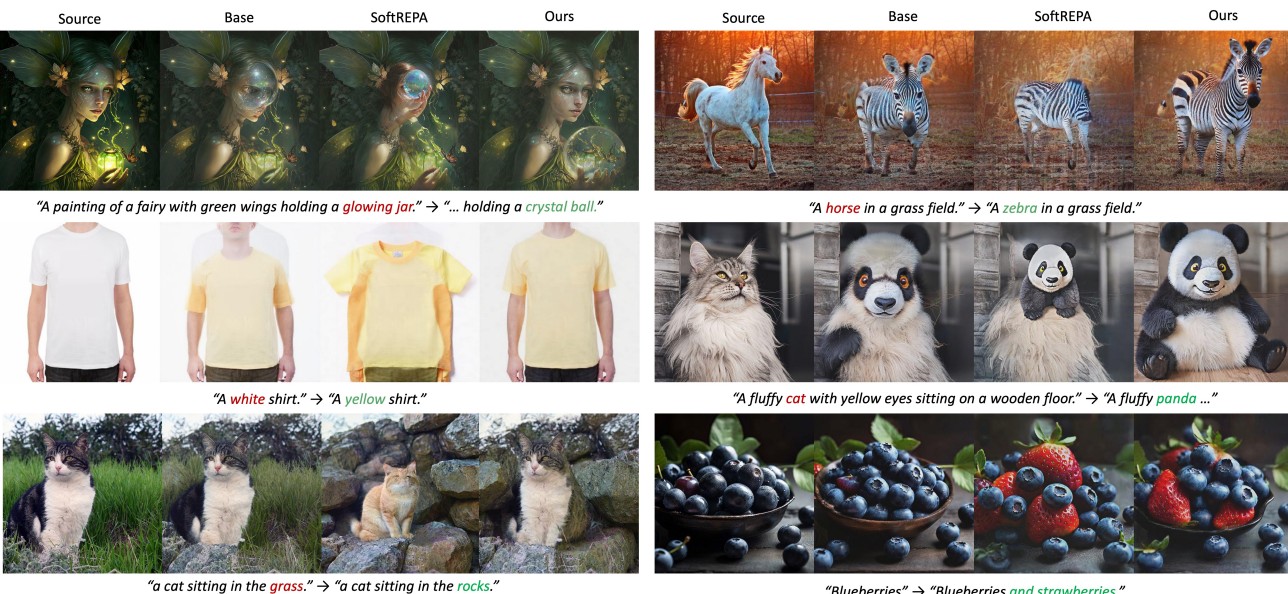

*Figure 6.* Qualitative comparison of image editing results from baseline methods, SoftREPA, and our method. The proposed method demonstrates a superior balance between text alignment and structural consistency.

As shown in Figure 5-(right), we evaluate performance from two complementary perspectives: text alignment (using CLIP similarity) and source consistency (using background LPIPS). While a trade-off exists in the two metrics, our method demonstrates a consistently superior balance, establishing a Pareto front compared to all baselines. While some baselines (*e.g.*, FlowEdit, FlowAlign, RF-Inversion) excel in source consistency, they struggle to achieve strong text alignment. Conversely, SoftREPA (Lee et al., 2025a) often achieve high CLIP similarity at the expense of source consistency. As shown qualitatively in Figure 6, SoftREPA (Lee et al., 2025a) often over-edits or generates artifacts, which distort the original image structure. Additionally, Table 2 provides detailed quantitative results, including human preference scores alongside text alignment and structural preservation metrics.

**Complementarity with Diffusion RL methods.** To further compare with other DPO-based methods and examine their complementarity with our approach, we conducted additional experiments by integrating our pretrained soft tokens into existing DPO frameworks. While DPO-based methods focus on preference alignment, our method targets representation alignment between text and image features by training soft text tokens to modulate the propagated features within a contrastive framework. To demonstrate the complementarity between the two paradigms, we combined the pretrained soft tokens with Diffusion-DPO (Wallace et al., 2024), SPO (Liang et al., 2025), and InPO (Lu et al., 2025) on SD1.5 and SDXL. As shown in Table 3, this simple integration consistently improves performance across all DPO-based baselines. The results suggest that combining explicit preference alignment with our representation-level text-image alignment provides a unified and more robust

| | Inversion | Method | Human Preference | | Text Alignment | | | Background Preservation | | |
|---|---|---|---|---|---|---|---|---|---|---|
| | | | Image -Reward↑ | Pick -Score↑ | CLIP/ Edited↑ | CLIP/ Whole↑ | HPSv2↑ | PSNR↑ | LPIPS/ Whole↓ | SSIM↑ |
| SD1.5 | ddim | MasaCtrl | -13.94 | 21.03 | 21.20 | 24.18 | 20.27 | **22.31** | 14.59 | **80.41** |
| | ddim | MasaCtrl + SoftREPA | -12.76 | **21.05** | 21.27 | 24.44 | **20.27** | 22.27 | 14.5 | 80.27 |
| | ddim | MasaCtrl + Ours | **-8.71** | 21.01 | **21.86** | **25.08** | 20.24 | 21.60 | **11.65** | 79.30 |
| | direct | MasaCtrl | 4.77 | 21.39 | 21.47 | 24.54 | **20.53** | **22.82** | 12.21 | **82.02** |
| | direct | MasaCtrl + SoftREPA | 4.48 | 21.40 | 21.49 | 24.69 | 20.52 | 22.77 | 12.19 | 81.85 |
| | direct | MasaCtrl + Ours | **7.79** | **21.41** | **22.19** | **25.53** | 20.52 | 21.99 | **9.87** | 80.78 |
| SD3 | - | RF-Inversion | 128.0 | 22.07 | 24.17 | 27.26 | **20.84** | **13.10** | **36.10** | 57.17 |
| | - | RF-Inversion + SoftREPA | 128.5 | 21.98 | 24.70 | 28.88 | 20.38 | 12.80 | 38.02 | 56.93 |
| | - | RF-Inversion + Ours | **132.3** | **22.13** | **24.72** | **29.07** | 20.58 | 12.90 | 37.17 | **57.98** |

*Table 2.* Quantitative evaluation of image editing performance of baseline, SoftREPA (Lee et al., 2025a) and our method on PnP (Tumanyan et al., 2023), MasaCtrl (Cao et al., 2023), and RF-Inversion (Rout et al., 2024). ImageReward, CLIP, HPS, LPIPS, and SSIM are scaled by $\times 10^2$ and Distance is scaled by $\times 10^3$.

| | Model | COCO val5K | | | | |
|---|---|---|---|---|---|---|
| | | ImageReward↑ | PickScore↑ | CLIP↑ | HPSv2↑ | FID↓ |
| SD1.5 | Ours | 34.50 | 21.59 | 27.23 | 25.66 | 25.94 |
| | DiffusionDPO (Wallace et al., 2024) | 29.09 | 21.65 | 26.52 | **26.46** | 27.85 |
| | DiffusionDPO (Wallace et al., 2024) + Ours | **42.47** | **21.79** | **27.34** | 26.28 | **27.27** |
| | SPO (Liang et al., 2025) | 18.98 | 21.58 | 25.83 | 26.51 | 33.76 |
| | SPO (Liang et al., 2025) + Ours | **34.86** | **21.94** | **26.53** | **26.92** | **30.67** |
| | InPO (Lu et al., 2025) | 62.12 | 21.83 | 27.01 | **28.80** | 34.47 |
| | InPO (Lu et al., 2025) + Ours | **67.95** | **22.02** | **27.37** | 28.33 | **33.67** |
| SDXL | Ours | 84.22 | 22.57 | 26.86 | 27.96 | 24.83 |
| | DiffusionDPO (Wallace et al., 2024) | 91.67 | 22.65 | **27.36** | 28.90 | 28.64 |
| | DiffusionDPO (Wallace et al., 2024) + Ours | **93.12** | **22.73** | 27.06 | **28.94** | **28.18** |
| | SPO (Liang et al., 2025) | 96.95 | 23.20 | 25.93 | **30.85** | 31.84 |
| | SPO (Liang et al., 2025) + Ours | **98.64** | **23.23** | **26.07** | 30.35 | **31.68** |
| | InPO (Lu et al., 2025) | 94.05 | 22.74 | 26.91 | **29.54** | 27.78 |
| | InPO (Lu et al., 2025) + Ours | **96.07** | **22.81** | **26.94** | 29.38 | **27.33** |

*Table 3.* Quantitative evaluation of comparison and complementarity with other Diffusion-RL methods on COCO-val5K dataset(Lin et al., 2014). ImageReward, CLIP, HPS, and LPIPS are scaled by $\times 10^2$.

| | Tokens | Data | ImageReward↑ | PickScore↑ | CLIP↑ | HPSv2↑ | FID↓ |
|---|---|---|---|---|---|---|---|
| (i) | $\psi^+$ | $\mathcal{D}^+$ | 94.79 | 22.26 | 26.93 | 27.81 | 34.46 |
| (ii) | shared $\psi$ | $\mathcal{D}^+, \mathcal{D}^-$ | 47.33 | 21.69 | 25.68 | 25.82 | **31.20** |
| (iii) | $\psi^+, \psi^-$ | $\mathcal{D}^+, \mathcal{D}^-$ | 103.3 | 22.39 | 27.00 | 28.22 | 34.08 |

*Table 4.* Ablation study on training strategy, including separated objectives over positive/negative subsets and soft token parameterization. In Tokens, shared $\psi$ uses a single token set for both $\mathcal{D}^+$ and $\mathcal{D}^-$.

| | Tokens | ImageReward↑ | PickScore↑ | CLIP↑ | HPSv2↑ | FID↓ |
|---|---|---|---|---|---|---|
| (i) | $\psi^+, \psi^-$ | 84.53 | 22.18 | 26.71 | 27.64 | 36.47 |
| (ii) | $\psi^+$ (Ours) | **103.3** | **22.39** | **27.00** | **28.22** | **34.08** |

*Table 5.* Ablation study on sampling strategy. (i) uses negative tokens for unconditonal prediction in CFG and (ii) samples only with positive tokens.

approach for enhancing text–image generation quality.

**Ablation Study on Training Strategy.** To isolate the effect of the positive/negative subset training ($\mathcal{D}^+, \mathcal{D}^-$) and the dual-token parameterization, we conduct a controlled ablation while keeping the total number of learnable soft tokens fixed. As shown in Table 4, we compare three variants: (i) training only positive soft tokens $\psi^+$ on $\mathcal{D}^+$; (ii) training on both $D^+$ and $D^-$ with shared soft tokens, without explicitly decoupling positive and negative guidance; and (iii) our AGSM, which uses separate soft tokens, $\psi^+$ and $\psi^-$, for the two subsets.

As shown in Table 4, the results indicate that using both $\mathcal{D}^+$ and $\mathcal{D}^-$ is critical for improving alignment metrics such as ImageReward, PickScore, CLIP, and HPSv2 scores.

However, when positive and negative guidance are optimized with shared soft tokens, the improvement comes at the cost of degraded generative quality. This confirms that AGSM's gains come from the combination of explicit positive/negative subset training and the structured dual-token design.

**Ablation Study on Sampling Strategy.** We examine sampling behavior by comparing two strategies: (i) sampling with only positive soft tokens ($\psi^+$) for both conditional and unconditional generation; (ii) sampling with positive soft tokens ($\psi^+$) for conditional prediction and negative soft tokens ($\psi^-$) for unconditional prediction. As illustrated in Table 5, sampling without negative tokens yields notably higher image quality, especially better in ImageReward and FID, suggesting that negative-token sampling can overly

| $\gamma^-$ | ImageReward↑ | PickScore↑ | CLIP↑ | HPSv2↑ | FID↓ |
|---|---|---|---|---|---|
| 1 | 96.71 | 22.36 | 26.78 | 28.22 | 35.21 |
| 0.5 | 98.44 | 22.31 | **27.02** | 27.96 | 33.88 |
| 0.1 (Ours) | **103.3** | 22.39 | 27.00 | **28.22** | 34.08 |
| 0.05 | 100.1 | **22.46** | 26.95 | 28.11 | **33.04** |
| 0 | 94.79 | 22.26 | 26.93 | 27.81 | 34.46 |

*Table 6.* Comparison on negative guidance scale on SD3, $\gamma^- \in \{0, 0.05, 0.1, 0.5, 1\}$. $\gamma^+$ is set to 1.

| | ImageReward↑ | PickScore↑ | CLIP↑ | HPSv2↑ | FID↓ |
|---|---|---|---|---|---|
| BT | 29.67 | 21.52 | 27.13 | 25.31 | **24.76** |
| PL (ours) | **34.50** | **21.59** | **27.23** | **25.66** | 25.94 |

*Table 7.* Comparison of loss objective between Bradley-Terry (BT) and Plackett-Luce(PL) model. BT consists of pairwise positive-negative components, and PT generalizes BT into multi-negative components.

suppress important visual information, degrading fidelity and diversity. We normalize all metrics to a consistent scale for comparison. Additional qualitative examples can be found in Appendix H.

**The Sensitivity on the Negative Guidance Scale.** We further analyze the sensitivity to the negative guidance scale $\gamma^-$. In practice, we use a larger value for models with stronger CFG (SD1.5, SDXL), and a smaller value for smaller CFG model (SD3). As shown in Table 6, SD3 remains stable across a range of scales, with $\gamma^-=0.1$ showing the best performance. After training, a fixed scale generalizes well across datasets and tasks at inference time without retraining.

**BT vs PT Loss Objective.** We further compare our PL-based multi-candidate formulation with a simpler pairwise Bradley–Terry (BT) alternative. BT performs pairwise positive–negative comparison, whereas PL naturally handles multiple in-batch negative prompts through normalized multi-candidate preference modeling. PL consistently improves all alignment metrics over BT, while BT gives a slightly lower FID. This supports the use of PL for multi-candidate alignment rather than reducing the objective to independent pairwise comparisons.

## 5. Related Works

Recent work has extended Direct Preference Optimization (DPO) to diffusion models. Diffusion-DPO (Wallace et al., 2024) adapts DPO (Rafailov et al., 2023) to text-conditioned diffusion processes, and several follow-up methods improve preference modeling in different ways. SPO (Liang et al., 2025) introduces step-aware preference signals during on-policy sampling. RankDPO (Karthik et al., 2024) generalizes preference learning to multi-sample ranking, while CaPO (Lee et al., 2025b) enhances fidelity by aggregating multiple reward signals. InPO (Lu et al., 2025) uses DDIM inversion to identify preference-relevant latent variables

for selective finetuning, and DSPO (Zhu et al., 2025) integrates preference learning into the score-matching objective. While these approaches focus on human preference optimization, our method instead targets intrinsic text–image representation alignment, offering a complementary direction to DPO-based RL finetuning.

Beyond preference-pair optimization, recent reward-based diffusion RL methods optimize text-to-image diffusion models using scalar feedback from external reward models (Liu et al., 2026; Zheng et al., 2025). DiffusionNFT (Zheng et al., 2025) adapts Negative-aware Fine-Tuning (NFT) (Chen et al., 2025) to diffusion models by using negative policy for policy optimization. Unlike DiffusionNFT, which defines implicit positive and negative samples through external rewards, AGSM derives them from intrinsic text–image representation alignment and injects the resulting guidance directly into score matching.

## 6. Conclusion

We introduced Alignment-Guided Score Matching, a training-light approach that fine-tunes soft tokens to enhance intrinsic text–image representation alignment in diffusion models. By replacing explicit contrastive objectives with a score-based formulation using PL preference model and training negative samples with explicit preference directions, our method stabilizes soft-token optimization and mitigates off-manifold divergence. Through extensive experiments on text-to-image generation and text-guided image editing, we demonstrate consistent gains in alignment quality across multiple diffusion backbones. Our approach is further shown to be complementary to existing DPO-based post-training methods, yielding additional improvements when combined with preference-optimization techniques. Ablation studies confirm the importance of utilizing negative samples on training and highlight the impact of sampling strategies involving negative tokens. Overall, this work provides a simple yet effective framework for strengthening text–image alignment within the diffusion training dynamics, offering a broadly applicable enhancement for modern generative models.

## Impact Statement

This paper presents work whose goal is to advance the field of Machine Learning. There are many potential societal consequences of our work, none which we feel must be specifically highlighted here.

## Acknowledgement

This work was supported by the National Research Foundation of Korea (NRF) grant funded by the Korea govern-

ment (MSIT) (RS-2026-25468886), the National Research Foundation of Korea under Grant RS-2024-00336454, the AI Computing Infrastructure Enhancement (GPU Rental Support) User Support Program funded by the Ministry of Science and ICT (MSIT), Republic of Korea (RQT-25-120217), the Advanced GPU Utilization Support Program funded by the Government of the Republic of Korea (Ministry of Science and ICT) (02-26-01-0404).

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

## A. Interpretation of Negative Target Distribution

A Bayes-consistent negative conditional distribution from Eq. (9) can be written as

$$p_t^-(\boldsymbol{x}_t|\boldsymbol{c}) := p_t(\boldsymbol{x}_t|\boldsymbol{c}, z=0) \propto p_t(\boldsymbol{x}_t|\boldsymbol{c})\, p(z=0|\boldsymbol{x}_t, \boldsymbol{c}). \tag{19}$$

Since $p(z=0|\boldsymbol{x}_t, \boldsymbol{c}) = 1 - p(z=1|\boldsymbol{x}_t, \boldsymbol{c})$, the corresponding guidance term becomes

$$\nabla_{\boldsymbol{x}_t} \log p(z=0|\boldsymbol{x}_t, \boldsymbol{c}) = \nabla_{\boldsymbol{x}_t} \log\left(1 - p(z=1|\boldsymbol{x}_t, \boldsymbol{c})\right) \tag{20}$$

$$= -\frac{\nabla_{\boldsymbol{x}_t} p(z=1|\boldsymbol{x}_t, \boldsymbol{c})}{1 - p(z=1|\boldsymbol{x}_t, \boldsymbol{c})}. \tag{21}$$

Instead of directly using $p(z=0|\boldsymbol{x}_t, \boldsymbol{c})$, we adopt the surrogate form $p(z=1|\boldsymbol{x}_t, \boldsymbol{c})^{-\gamma^-}$, which yields

$$\nabla_{\boldsymbol{x}_t} \log p(z=1|\boldsymbol{x}_t, \boldsymbol{c})^{-\gamma^-} = -\gamma^- \nabla_{\boldsymbol{x}_t} \log p(z=1|\boldsymbol{x}_t, \boldsymbol{c}) \tag{22}$$

$$= -\gamma^- \frac{\nabla_{\boldsymbol{x}_t} p(z=1|\boldsymbol{x}_t, \boldsymbol{c})}{p(z=1|\boldsymbol{x}_t, \boldsymbol{c})}. \tag{23}$$

Therefore,

$$\nabla_{\boldsymbol{x}_t} \log p(z=0|\boldsymbol{x}_t, \boldsymbol{c}) \parallel \nabla_{\boldsymbol{x}_t} \log p(z=1|\boldsymbol{x}_t, \boldsymbol{c})^{-\gamma^-}. \tag{24}$$

That is, both gradients of Eq. (20) and Eq. (22) are pointwise proportional with a positive scaling factor and share the same directional component $-\nabla_{\boldsymbol{x}_t} p(z=1|\boldsymbol{x}_t, \boldsymbol{c})$.

Therefore, both formulations induce repulsive updates away from regions with high alignment probability, differing only in their local scaling factors.

Accordingly, the proposed inverse weighting can be interpreted as a surrogate negative guidance term that preserves the repulsive direction of the Bayes-consistent formulation while yielding a unified additive score form.

## B. Reward Function Derivation

We derive the implicit reward used in our alignment objective. Starting from the definition in Equation (8), we define the reward as the expected log-likelihood under the DDPM posterior:

$$r(\boldsymbol{x}_t, \boldsymbol{c}) := \lambda_t\, \mathbb{E}_{q(\boldsymbol{x}_{t-1}|\boldsymbol{x}_t, \boldsymbol{x}_0)}[\log p_\theta(\boldsymbol{x}_{t-1} \mid \boldsymbol{x}_t, \boldsymbol{c})]. \tag{25}$$

The reverse process of DDPM (Ho et al., 2020) provides an explicit Gaussian parameterization for $p_\theta(\boldsymbol{x}_{t-1} \mid \boldsymbol{x}_t, \boldsymbol{c})$:

$$p_\theta(\boldsymbol{x}_{t-1} \mid \boldsymbol{x}_t, \boldsymbol{c}) = \mathcal{N}\left(\boldsymbol{x}_{t-1};\, \mu_\theta(\boldsymbol{x}_t, t, \boldsymbol{c}), \sigma_t^2 \mathbf{I}\right), \tag{26}$$

where $\mu_\theta(\boldsymbol{x}_t, t, \boldsymbol{c}) = \frac{1}{\sqrt{\alpha_t}}\left(\boldsymbol{x}_t - \frac{\beta_t}{\sqrt{1-\bar{\alpha}_t}} \boldsymbol{\epsilon}_\theta(\boldsymbol{x}_t, t, \boldsymbol{c})\right)$, and $\sigma_t^2 = \frac{1-\bar{\alpha}_{t-1}}{1-\bar{\alpha}_t}\beta_t$, $\alpha_t = 1 - \beta_t$, $\bar{\alpha}_t = \prod_{s=1}^t \alpha_s$. Since both $q(\boldsymbol{x}_{t-1} \mid \boldsymbol{x}_t, \boldsymbol{x}_0)$ and $p_\theta(\boldsymbol{x}_{t-1} \mid \boldsymbol{x}_t, \boldsymbol{c})$ are Gaussian, expanding the log-density yields

$$r(\boldsymbol{x}_t, \boldsymbol{c}) = -\frac{\lambda_t}{2\sigma_t^2}\mathbb{E}_q\left[\|\boldsymbol{x}_{t-1} - \mu_\theta\|_2^2\right] + C_1 \tag{27}$$

where $C_1$ is a timestep dependent constant. Using

$$\mathbb{E}\|\boldsymbol{x}_{t-1} - \mu_\theta\|^2 = \|\tilde{\mu}_t - \mu_\theta\|^2 + d\,\tilde{\beta}_t \tag{28}$$

with $\tilde{\mu}_t = \mathbb{E}[\boldsymbol{x}_{t-1} \mid \boldsymbol{x}_t, \boldsymbol{x}_0]$ and posterior variance $\tilde{\beta}_t$,

$$r(\boldsymbol{x}_t, \boldsymbol{c}) = -\frac{\lambda_t}{2\sigma_t^2}\|\tilde{\mu}_t - \mu_\theta\|_2^2 + C_2. \tag{29}$$

Since $C_2$ is independent of $\theta$ and $\boldsymbol{x}_t$, expressing both $\tilde{\mu}_t$ and $\mu_\theta$ in the noise parameterization gives

$$r(\boldsymbol{x}_t, \boldsymbol{c}) = -\frac{\lambda_t \beta_t}{2\alpha_t(1-\bar{\alpha}_{t-1})} \left\|\boldsymbol{\epsilon}_\theta^{\hat{\psi}_z}(\boldsymbol{x}_t, t, \boldsymbol{c}) - \boldsymbol{\epsilon}_t\right\|_2^2 \tag{30}$$

EMA soft tokens $\hat{\psi}_z$ are used to stabilize the reward evaluation. Thus, higher reward corresponds to closer agreement between model-predicted noise and the forward noise realization.

## C. Alignment-Guided Score Matching Derivation

To compute the gradient needed in the alignment score model (Equation (13)), we differentiate the reward w.r.t. $\boldsymbol{x}_t$:

$$\nabla_{\boldsymbol{x}_t} r(\boldsymbol{x}_t, \boldsymbol{c}) = -\tfrac{\lambda_t \beta_t}{\alpha_t (1 - \bar{\alpha}_{t-1})} \mathbf{J}_{\boldsymbol{\epsilon}_\theta^{\hat{\psi}_z}}(\boldsymbol{x}_t)^\top \big(\boldsymbol{\epsilon}_\theta^{\hat{\psi}_z}(\boldsymbol{x}_t, t, \boldsymbol{c}) - \boldsymbol{\epsilon}_t\big), \tag{31}$$

where we omit computing the jacobian $\mathbf{J}_{\boldsymbol{\epsilon}_\theta^{\hat{\psi}_z}}(\boldsymbol{x}_t) = \tfrac{\partial \boldsymbol{\epsilon}_\theta^{\hat{\psi}_z}(\boldsymbol{x}_t, t, \boldsymbol{c})}{\partial \boldsymbol{x}_t}$. Plugging Equation (31) into Equation (13), the gradient of the alignment score model becomes

$$\nabla_{\boldsymbol{x}_t} \log p(z{=}1 \mid \boldsymbol{x}_t, \boldsymbol{c}) = -A(t)\Big(\boldsymbol{\epsilon}_\theta^{\hat{\psi}_z}(\boldsymbol{x}_t, t, \boldsymbol{c}) - \boldsymbol{\epsilon}_t - \sum_i w_i \,(\boldsymbol{\epsilon}_\theta^{\hat{\psi}_z}(\boldsymbol{x}_t, t, \boldsymbol{c}^i) - \boldsymbol{\epsilon}_t)\Big) \tag{32}$$

$$= -A(t)\Big(\boldsymbol{\epsilon}_\theta^{\hat{\psi}_z}(\boldsymbol{x}_t, t, \boldsymbol{c}) - \sum_i w_i \,\boldsymbol{\epsilon}_\theta^{\hat{\psi}_z}(\boldsymbol{x}_t, t, \boldsymbol{c}^i)\Big), \tag{33}$$

where $A(t) = \tfrac{\lambda_t \beta_t}{\alpha_t (1 - \bar{\alpha}_{t-1})}$. With the same realization of $\boldsymbol{\epsilon}_t$, $\boldsymbol{\epsilon}_t$ cancels between positive and negative terms, giving a clean contrast between the positive prediction and the weighted average. Using the definition of the score function which connects the score model and diffusion models described in (Song et al., 2021b), we can derive $\nabla_{\boldsymbol{x}_t} \log \tfrac{p_\theta(\boldsymbol{x}_t | \boldsymbol{c})}{p_{data}(\boldsymbol{x}_t | \boldsymbol{c})} = -\tfrac{1}{\sqrt{1 - \bar{\alpha}_t}}(\boldsymbol{\epsilon}_\theta(\boldsymbol{x}_t, \boldsymbol{c}, t) - \boldsymbol{\epsilon}_t)$. By combining this and Equation (11), Equation (12) becomes

$$\mathcal{L}(\psi_z) = \mathbb{E}_{\substack{(\boldsymbol{x}, \boldsymbol{c}) \sim D, \\ t \sim U(0,1), \\ \boldsymbol{\epsilon} \sim \mathcal{N}(0, \mathbf{I})}}\Big[ w(t) \| \nabla \log p_{t,\theta}^{\psi_z}(\boldsymbol{x}_t | \boldsymbol{c}) - \nabla \log \tilde{p}_t(\boldsymbol{x}_t | \boldsymbol{c}, z) \|_2^2 \Big] \tag{34}$$

$$= \mathbb{E}_{\substack{(\boldsymbol{x}, \boldsymbol{c}) \sim D, \\ t \sim U(0,1), \\ \boldsymbol{\epsilon} \sim \mathcal{N}(0, \mathbf{I})}}\Big[ w(t) \| \nabla \log p_{t,\theta}^{\psi_z}(\boldsymbol{x}_t | \boldsymbol{c}) - (\nabla \log p_t(\boldsymbol{x}_t | \boldsymbol{c}) + \gamma_z \nabla \log p(z{=}1 \mid \boldsymbol{x}_t, \boldsymbol{c})) \|_2^2 \Big] \tag{35}$$

$$= \mathbb{E}_{\substack{(\boldsymbol{x}, \boldsymbol{c}) \sim D, \\ t \sim U(0,1), \\ \boldsymbol{\epsilon} \sim \mathcal{N}(0, \mathbf{I})}}\Big[ w(t) \| - \tfrac{1}{\sqrt{1 - \bar{\alpha}_t}} \boldsymbol{\epsilon}_\theta^{\psi_z}(\boldsymbol{x}_t, t, \boldsymbol{c}) - (-\tfrac{1}{\sqrt{1 - \bar{\alpha}_t}} \boldsymbol{\epsilon}_t + \gamma_z \nabla \log p(z{=}1 \mid \boldsymbol{x}_t, \boldsymbol{c})) \|_2^2 \Big]. \tag{36}$$

By leveraging the gradient of the alignment score model Eq. (33), the final score matching loss can be rewritten as follows:

$$\mathcal{L}(\psi_z) = \mathbb{E}_{\substack{(\boldsymbol{x}, \boldsymbol{c}) \sim D, \\ t \sim U(0,1), \\ \boldsymbol{\epsilon} \sim \mathcal{N}(0, \mathbf{I})}} \big[ \tfrac{w(t)}{1 - \bar{\alpha}_t} \| \boldsymbol{\epsilon}_\theta^{\psi_z}(\boldsymbol{x}_t, t, \boldsymbol{c}) - (\boldsymbol{\epsilon}_t + \tfrac{\gamma_z \lambda_t \beta_t \sqrt{1 - \bar{\alpha}_t}}{\alpha_t (1 - \bar{\alpha}_{t-1})}(\boldsymbol{\epsilon}_\theta^{\hat{\psi}_z}(\boldsymbol{x}_t, t, \boldsymbol{c}) - \sum_i w_i \boldsymbol{\epsilon}_\theta^{\hat{\psi}_z}(\boldsymbol{x}_t, t, \boldsymbol{c}^i))) \|_2^2 \big]$$

$$= \mathbb{E}_{\substack{(\boldsymbol{x}, \boldsymbol{c}) \sim D, \\ t \sim U(0,1), \\ \boldsymbol{\epsilon} \sim \mathcal{N}(0, \mathbf{I})}} \big[ \tilde{w}(t) \| \boldsymbol{\epsilon}_\theta^{\psi_z}(\boldsymbol{x}_t, t, \boldsymbol{c}) - (\boldsymbol{\epsilon}_t + \gamma_z \tilde{A}(t)(\boldsymbol{\epsilon}_\theta^{\hat{\psi}_z}(\boldsymbol{x}_t, t, \boldsymbol{c}) - \sum_i w_i \boldsymbol{\epsilon}_\theta^{\hat{\psi}_z}(\boldsymbol{x}_t, t, \boldsymbol{c}^i))) \|_2^2 \big] \tag{37}$$

where $\tilde{A}(t) = \tfrac{\lambda_t \beta_t \sqrt{1 - \bar{\alpha}_t}}{\alpha_t (1 - \bar{\alpha}_{t-1})}$ and $\tilde{w}(t) = \tfrac{w(t)}{1 - \bar{\alpha}_t}$.

## D. Alignment-Guided Score Matching in Flow Model

**Flow model.** Flow model defines the interpolant $\boldsymbol{x}_t$ between data $\boldsymbol{x}_0 \sim p(\boldsymbol{x})$ and noise $\boldsymbol{\epsilon} \sim \mathcal{N}(\mathbf{0}, \mathbf{I})$ as

$$\boldsymbol{x}_t = (1 - t)\boldsymbol{x}_0 + t\,\boldsymbol{\epsilon}, \qquad t \in [0, 1], \tag{38}$$

and trains the flow model $v_\theta$ to match the target velocity via

$$\mathcal{L}_{\text{flow}} = \mathbb{E}_{\boldsymbol{x}_0, \boldsymbol{\epsilon}, t, \boldsymbol{c}}\big[ \| v_\theta(\boldsymbol{x}_t, t, \boldsymbol{c}) - (\boldsymbol{\epsilon} - \boldsymbol{x}_0) \|_2^2 \big]. \tag{39}$$

**Text-image alignment in flow model.** In the reward fitting process, we measure the alignment between an image and a text pair $(\boldsymbol{x}_t, \boldsymbol{c})$, denoted as $z$, using a Plackett-Luce (PL) model:

$$p(z | \boldsymbol{x}_t, \boldsymbol{c}) = \frac{\exp\big(r(\boldsymbol{x}_t, \boldsymbol{c})\big)}{\sum_i \exp\big(r(\boldsymbol{x}_t, \boldsymbol{c}^i)\big)}, \qquad w_i = \frac{\exp\big(r(\boldsymbol{x}_t, \boldsymbol{c}^i)\big)}{\sum_j \exp\big(r(\boldsymbol{x}_t, \boldsymbol{c}^j)\big)}. \tag{40}$$

Following SoftREPA (Lee et al., 2025a), which interprets the negative denoising score-matching loss as a logit of contrastive learning, we extend this idea to the flow model by defining the reward using the flow model's conditional likelihood:

$$r(\boldsymbol{x}_t, \boldsymbol{c}) = \lambda_t \log p_\theta(\boldsymbol{x}_t \mid \boldsymbol{x}_{t+\Delta}, \boldsymbol{c}), \qquad p_\theta = \mathcal{N}(\boldsymbol{x}_t; \, \mu_\theta, \, \sigma_{t+\Delta}^2 \mathbf{I}), \quad \mu_\theta = \boldsymbol{x}_{t+\Delta} - \Delta \, v_\theta(\boldsymbol{x}_{t+\Delta}, t+\Delta, \boldsymbol{c}). \tag{41}$$

Here $\Delta > 0$ denotes a small time step, and $\sigma_{t+\Delta}^2$ represents the local transition variance (e.g., from the underlying probability–flow ODE). Under this local approximation, the flow dynamics

$$\frac{d\boldsymbol{x}_t}{dt} = v_\theta(\boldsymbol{x}_t, t, \boldsymbol{c}) \tag{42}$$

can be locally approximated (via first-order Euler discretization) as a Gaussian transition:

$$p_\theta(\boldsymbol{x}_t \mid \boldsymbol{x}_{t+\Delta}, \boldsymbol{c}) \simeq \mathcal{N}(\boldsymbol{x}_t; \, \boldsymbol{x}_{t+\Delta} - \Delta \, v_\theta(\boldsymbol{x}_{t+\Delta}, t+\Delta, \boldsymbol{c}), \, \sigma_{t+\Delta}^2 \mathbf{I}), \tag{43}$$

which describes the probability of reaching $\boldsymbol{x}_t$ from $\boldsymbol{x}_{t+\Delta}$ under the flow model $v_\theta$. Then, the estimated reward becomes

$$r(\boldsymbol{x}_t, \boldsymbol{c}) = -\frac{\lambda_t}{2\,\sigma_{t+\Delta}^2} \left\| \boldsymbol{x}_t - \mu_\theta^{\hat{\psi}_z} \right\|_2^2, \qquad \mu_\theta^{\hat{\psi}_z} = \boldsymbol{x}_{t+\Delta} - \Delta \, v_\theta^{\hat{\psi}_z}(\boldsymbol{x}_{t+\Delta}, t+\Delta, \boldsymbol{c}). \tag{44}$$

To further stabilize the reward calculation, we used flow model with soft tokens updated by exponential moving average (EMA), $\hat{\psi}_z$. The gradient of the reward with respect to $\boldsymbol{x}_t$ is

$$\nabla_{\boldsymbol{x}_t} r(\boldsymbol{x}_t, \boldsymbol{c}) = -\frac{\lambda_t}{\sigma_{t+\Delta}^2} (\boldsymbol{x}_t - \mu_\theta^{\hat{\psi}_z}). \tag{45}$$

Plugging Equation (45) into Equation (40), we obtain

$$\begin{aligned} \nabla_{\boldsymbol{x}_t} \log p(z{=}1|\boldsymbol{x}_t, \boldsymbol{c}) &= \nabla_{\boldsymbol{x}_t} r(\boldsymbol{x}_t, \boldsymbol{c}) - \sum_i w_i \, \nabla_{\boldsymbol{x}_t} r(\boldsymbol{x}_t, \boldsymbol{c}^i) \\ &= -\frac{\lambda_t}{\sigma_{t+\Delta}^2}(\boldsymbol{x}_t - \mu_\theta^{\hat{\psi}_z}(\boldsymbol{c})) + \sum_i w_i \frac{\lambda_t}{\sigma_{t+\Delta}^2}(\boldsymbol{x}_t - \mu_\theta^{\hat{\psi}_z}(\boldsymbol{c}^i)) \\ &= \frac{\lambda_t}{\sigma_{t+\Delta}^2}\left(\mu_{\theta'}(\boldsymbol{c}) - \sum_i w_i \, \mu_\theta^{\hat{\psi}_z}(\boldsymbol{c}^i)\right). \end{aligned} \tag{46}$$

Substituting $\mu_\theta^{\hat{\psi}_z}(\boldsymbol{c}) = \boldsymbol{x}_{t+\Delta} - \Delta \, v_\theta^{\hat{\psi}_z}(\boldsymbol{x}_{t+\Delta}, t+\Delta, \boldsymbol{c})$ cancels out $\boldsymbol{x}_{t+\Delta}$ and yields

$$\nabla_{\boldsymbol{x}_t} \log p(z{=}1 \mid \boldsymbol{x}_t, \boldsymbol{c}) = -\frac{\lambda_t \Delta}{\sigma_{t+\Delta}^2}\left(v_\theta^{\hat{\psi}_z}(\boldsymbol{x}_{t+\Delta}, t+\Delta, \boldsymbol{c}) - \sum_i w_i \, v_\theta^{\hat{\psi}_z}(\boldsymbol{x}_{t+\Delta}, t+\Delta, \boldsymbol{c}^i)\right). \tag{47}$$

**Score matching loss in flow model.** Starting from the score matching objective

$$\mathcal{L}(\psi_z) = \mathbb{E}_{\substack{(\boldsymbol{x}, \boldsymbol{c}) \sim D, \\ t \sim U(0,1), \\ \boldsymbol{\epsilon} \sim \mathcal{N}(0, \mathbf{I})}} \left[ w(t) \left\| \nabla_{\boldsymbol{x}_t} \log p_{t,\theta}^{\psi_z}(\boldsymbol{x}_t \mid \boldsymbol{c}) - \left( \nabla_{\boldsymbol{x}_t} \log p_t(\boldsymbol{x}_t \mid \boldsymbol{c}) + \gamma_z \, \nabla_{\boldsymbol{x}_t} \log p(z{=}1 \mid \boldsymbol{x}_t, \boldsymbol{c}) \right) \right\|_2^2 \right], \tag{48}$$

we use the probability–flow ODE relation

$$v(\boldsymbol{x}_t, t, \boldsymbol{c}) = -K(t) \, \nabla_{\boldsymbol{x}_t} \log p_t(\boldsymbol{x}_t \mid \boldsymbol{c}), \qquad K(t) > 0, \tag{49}$$

where $K(t)$ is a time-dependent scaling factor determined by the flow formulation (e.g., $K(t) = \frac{1}{2}g(t)^2$ in a probability–flow ODE).

Substituting

$$v_{t,\theta}^{\psi_z}(\boldsymbol{x}_t, \boldsymbol{c}) := -K(t) \, \nabla_{\boldsymbol{x}_t} \log p_{t,\theta}^{\psi_z}(\boldsymbol{x}_t \mid \boldsymbol{c}),$$

and the gradient of alignment score model

$$\nabla_{\boldsymbol{x}_t} \log p(z{=}1 \mid \boldsymbol{x}_t, \boldsymbol{c}) = -\frac{\lambda_t \Delta}{\sigma_{t+\Delta}^2} \Big( v_\theta^{\hat{\psi}_z}(\boldsymbol{x}_{t+\Delta}, t{+}\Delta, \boldsymbol{c}) - \sum_i w_i\, v_\theta^{\hat{\psi}_z}(\boldsymbol{x}_{t+\Delta}, t{+}\Delta, \boldsymbol{c}^i) \Big),$$

into the score matching objective, we obtain

$$\mathcal{L}(\psi_z) = \mathbb{E}\Big[ w(t) \big\| \nabla_{\boldsymbol{x}_t} \log p_{t,\theta}^{\psi_z}(\boldsymbol{x}_t \mid \boldsymbol{c}) - \big( \nabla_{\boldsymbol{x}_t} \log p_t(\boldsymbol{x}_t \mid \boldsymbol{c}) + \gamma_z \nabla_{\boldsymbol{x}_t} \log p(z{=}1 \mid \boldsymbol{x}_t, \boldsymbol{c}) \big) \big\|_2^2 \Big]$$

$$= \mathbb{E}\Big[ \frac{w(t)}{K(t)^2} \big\| v_{t,\theta}^{\psi_z}(\boldsymbol{x}_t, \boldsymbol{c}) - \big( v_t(\boldsymbol{x}_t, \boldsymbol{c}) + \gamma_z K(t) \frac{\lambda_t \Delta}{\sigma_{t+\Delta}^2} \big( v_\theta^{\hat{\psi}_z}(\boldsymbol{x}_{t+\Delta}, t{+}\Delta, \boldsymbol{c}) - \sum_i w_i\, v_\theta^{\hat{\psi}_z}(\boldsymbol{x}_{t+\Delta}, t{+}\Delta, \boldsymbol{c}^i) \big) \big) \big\|_2^2 \Big]$$

$$(50)$$

where the expectation is over $(\boldsymbol{x}, \boldsymbol{c}, z) \sim D$, $t \sim U(0,1)$, and $\epsilon \sim \mathcal{N}(0, \mathbf{I})$.

Finally, redefining weighting term as $\tilde{w}(t) = w(t)/K(t)^2$ and $B(t) = K(t) \frac{\lambda_t \Delta}{\sigma_{t+\Delta}^2}$ the loss can be expressed as

$$\mathcal{L}(\psi_z) = \mathbb{E}\Big[ \tilde{w}(t) \big\| v_{t,\theta}^{\psi_z}(\boldsymbol{x}_t, \boldsymbol{c}) - \big( v_t(\boldsymbol{x}_t, \boldsymbol{c}) + \gamma_z B(t) \big( v_\theta^{\hat{\psi}_z}(\boldsymbol{x}_{t+\Delta}, t{+}\Delta, \boldsymbol{c}) - \sum_i w_i\, v_\theta^{\hat{\psi}_z}(\boldsymbol{x}_{t+\Delta}, t{+}\Delta, \boldsymbol{c}^i) \big) \big) \big\|_2^2 \Big], \quad (51)$$

which mirrors the diffusion-based objective in Equation (37) while entirely derived within the flow-based formulation.

# E. Additional Results on Image Generation

**COCO-val T2I Generation.** In this section, we present detailed quantitative results on the COCO-val image generation task, comparing the baseline, SoftREPA, and our method across different backbones. As shown in Table 8, our method consistently outperforms the baselines in both human preference scores and CLIP scores. With respect to the trade-off between human preference scores and FID, our method achieves lower FID than SoftREPA while maintaining strong preference-aligned performance.

| | Model | ImageReward↑ | PickScore↑ | CLIP↑ | HPSv2↑ | FID↓ |
|---|---|---|---|---|---|---|
| | | **COCO val5K** | | | | |
| SD1.5 | SD1.5 | 17.72 | 21.47 | 26.4 | 25.08 | **24.59** |
| | SoftREPA | **40.02** | **21.64** | 27.09 | **26.05** | 29.25 |
| | Ours | 34.50 | 21.59 | **27.23** | 25.66 | 25.94 |
| SDXL | SDXL | 75.06 | 22.38 | 26.76 | 27.35 | **24.69** |
| | SoftREPA | **85.28** | **22.63** | 26.78 | **28.41** | 26.42 |
| | Ours | 84.22 | 22.57 | **26.86** | 27.96 | 24.83 |
| SD3 | SD3 | 94.27 | 22.54 | 26.30 | 28.09 | **31.59** |
| | SoftREPA | **108.5** | **22.55** | 26.91 | **28.91** | 36.21 |
| | Ours | 103.3 | 22.39 | **27.00** | 28.22 | 34.08 |

*Table 8.* Quantitative evaluation of T2I generation on SD1.5, SDXL, and SD3. Generation quality is evaluated on the COCO-val 5K (Lin et al., 2014) and GenEval (Ghosh et al., 2023) benchmark. ImageReward, CLIP, HPS, and LPIPS are scaled by $\times 10^2$.

**Long-prompt T2I Generation.** To further evaluate generalization to long-text prompts, we additionally test on UniGen-Bench++ (Wang et al., 2026) with 600 long-text T2I prompts, comparing SD3, SoftREPA, and our method. As shown in Table 9, our method achieves the best ImageReward, CLIP, PickScore, and HPSv2, indicating that the proposed method remains effective beyond short COCO-style captions.

# F. Additional Results on Image Editing

In this section, we provide implementation details on the baseline methods used in the image editing experiments. We provide quantitative evaluation results of all editing methods in Table 10. Our method enhances text alignment of baseline methods with comparable or superior background preservation. In Figure 7, we present additional qualitative comparison on baseline editing methods.

| Model | ImageReward | CLIP | PickScore | HPSv2 |
|---|---|---|---|---|
| SD3 Base | 82.33 | 29.50 | 21.32 | 28.87 |
| SoftREPA | 90.63 | 28.93 | 21.18 | 28.61 |
| Ours | **98.01** | **29.56** | **21.48** | **29.66** |

*Table 9.* Comparison on UniGenBench++ (Wang et al., 2026).

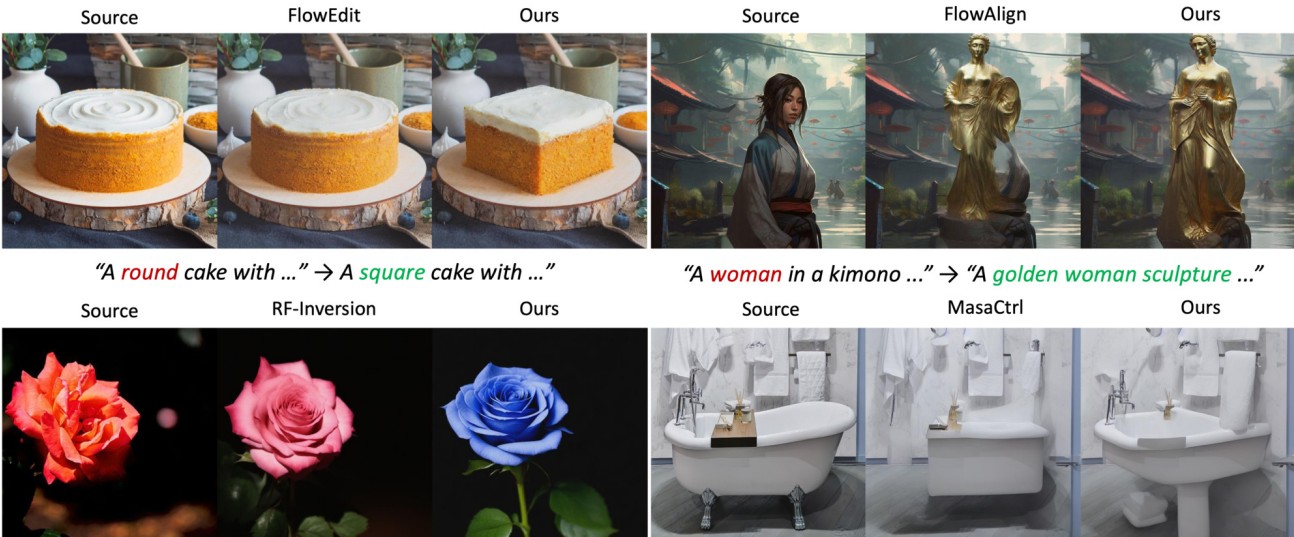

*Figure 7.* Qualitative comparison of our proposed method with different editing methods.

## Baseline methods - SD3

1. RF-Inversion (Rout et al., 2024) : We employ RF-Inversion as a representative baseline for image editing utilizing the inversion process within the SD3 framework. Based on the official implementation, we configure the parameters as follows: $\gamma = 0.5$, $\eta = 0.9$, a starting time $s = 0$, and a stopping time $\tau = 0.25$. Null-text embedding is leveraged for the inversion stage, and a Classifier-Free Guidance (CFG) scale of 13.5 is applied during the sampling phase to ensure consistency with the FlowEdit implementation.

2. FlowEdit (Kulikov et al., 2024) : FlowEdit is selected as a baseline method that effectively bypasses the inversion process in SD3. We utilize a CFG scale of 3.5 for the source direction and 13.5 for the target direction. Additionally, the starting index of timestep is set to 18 out of a total of 50 timesteps, resulting in 33 timesteps.

3. FlowAlign (Kim et al., 2025) : We include FlowAlign, a method that improves FlowEdit by regularizing the editing trajectory, demonstrating superior source consistency. Consistent with the FlowEdit configuration, the target CFG scale is set to 13.5. We adopt the official implementation's setting for the regularization coefficient, $\zeta = 0.01$.

## Baseline methods - SD1.5

1. MasaCtrl (Cao et al., 2023) : For an editing method that adapts the self-attention mechanism to enable consistent synthesis, we select MasaCtrl. Following the original work, we configure the initial layout synthesis to stop at step $S = 4$ and initiate the mutual self-attention control from layer $L = 10$. We evaluate this method using both DDIM inversion and a direct inversion approach. NFE is set to 50 with a CFG scale of 7.0.

2. PnP (Tumanyan et al., 2023) : PnP performs text-driven image-to-image translation by extracting and injecting spatial features ($f$) from intermediate decoder layers and self-attention maps ($A$) from the guidance image's generation into the target image's generation. For our experiments, which use 50 total sampling steps, we set the injection thresholds as $\tau_A = 25$ and $\tau_f = 40$, consistent with the official implementation. PnP is implemented on both DDIM and direct inversion with a CFG scale of 7.0.

| | Inversion | Method | Human Preference | | Text Alignment | | | Background Preservation | | |
|---|---|---|---|---|---|---|---|---|---|---|
| | | | Image-Reward↑ | Pick-Score↑ | CLIP/Edited↑ | CLIP/Whole↑ | HPS↑ | PSNR↑ | LPIPS/Whole↓ | SSIM↑ |
| **SD1.5** | ddim | PnP | **32.29** | **21.53** | 22.56 | 25.59 | **20.55** | 22.31 | 14.48 | 79.58 |
| | ddim | PnP + Ours | 26.74 | 21.49 | **22.95** | **26.10** | 20.52 | **22.57** | **10.81** | **80.14** |
| | direct | PnP | **40.85** | **21.65** | 22.68 | 25.64 | 20.61 | 22.46 | 13.44 | 80.22 |
| | direct | PnP + Ours | 36.59 | 21.62 | **22.99** | **26.18** | **20.62** | **22.74** | **10.08** | **80.70** |
| | ddim | MasaCtrl | -13.94 | **21.03** | 21.20 | 24.18 | **20.27** | 22.31 | 14.59 | **80.41** |
| | ddim | MasaCtrl + Ours | **-8.71** | 21.01 | **21.86** | **25.08** | 20.24 | 21.60 | **11.65** | 79.30 |
| | direct | MasaCtrl | 4.77 | 21.39 | 21.47 | 24.54 | **20.53** | 22.82 | 12.21 | **82.02** |
| | direct | MasaCtrl + Ours | **7.79** | **21.41** | **22.19** | **25.53** | 20.52 | 21.99 | **9.87** | 80.78 |
| **SD3** | o | RF-Inversion | 128.0 | 22.07 | 24.17 | 27.26 | **20.84** | **13.10** | **36.10** | 57.17 |
| | o | RF-Inversion + Ours | **132.3** | **22.13** | **24.72** | **29.07** | 20.58 | 12.90 | 37.17 | **57.98** |
| | x | FlowEdit | 103.0 | 22.08 | 23.35 | 26.46 | **21.11** | **21.44** | **10.84** | **81.36** |
| | x | FlowEdit + Ours | **114.8** | **22.36** | **24.59** | **28.47** | 21.02 | 20.20 | 13.86 | 78.54 |
| | x | FlowAlign | 68.00 | 21.50 | 22.38 | 25.65 | **20.85** | **24.21** | **6.89** | **86.44** |
| | x | FlowAlign + Ours | **81.97** | **21.63** | **23.33** | **27.40** | 20.67 | 22.65 | 9.58 | 83.95 |

*Table 10.* Quantitative evaluation of image editing performance our method compared to baseline methods. ImageReward, CLIP, HPS, LPIPS, and SSIM are scaled by $\times 10^2$ and Distance is scaled by $\times 10^3$.

## G. Implementation Details

All experiments were performed on two NVIDIA A100 GPUs, and detailed training configurations can be found in Table 11, using a configuration almost identical to that of SoftREPA (Lee et al., 2025a). For inference, we used positive soft tokens ($\psi^+$) for both conditional and unconditional generation. In Appendix H, additional results for using both positive and negative soft tokens together are provided.

| Models | lr | wd | total batch size (positive:negative) | iterations | token init | optimizer | lr scheduler |
|---|---|---|---|---|---|---|---|
| SD1.5 | 1e-3 | 1e-4 | 16(1 : 3) | 100,000 | $\varnothing$ | AdamW | CosineAnnealingWarmRestarts |
| SDXL | 1e-3 | 1e-4 | 16(1 : 3) | 1,000 | $N(0, 0.02)$ | AdamW | CosineAnnealingWarmRestarts |
| SD3 | 1e-3 | 1e-4 | 16(1 : 3) | 100,000 | $N(0, 0.02)$ | AdamW | CosineAnnealingWarmRestarts |

*Table 11.* The implementation details for training.

## H. Additional Results on Ablation Studies

**The Number of Soft Tokens.** We also study the effect of the number and type of optimized soft tokens. As shown in Table 12, using eight text tokens does not improve performance, consistent with SoftREPA's observation that performance is stable around four tokens and can degrade with more tokens. Interestingly, optimizing both text and image tokens remains effective in our framework, demonstrating flexibility of our proposed method.

**The Effect of Batch Size.** We further analyze the effect of batch size, which determines the size of the in-batch negative prompt pool. As shown in Table 13, increasing the batch size from 4 to 16 improves alignment-related metrics, while increasing it further to 64 yields only marginal gains. This suggests that a larger negative pool is useful up to a point, but our method does not rely on very large batch sizes.

**Sampling Strategy on Negative Tokens.** To evaluate the role of negative soft tokens, we compare alternative sampling strategies during inference. Our default configuration uses only positive soft tokens for both the conditional and unconditional predictions in classifier-free guidance (CFG). To assess whether negative soft tokens can further suppress the modeling of complement set or counterfactual concepts of given prompt, we also test a variant where the conditional prediction uses positive soft tokens and the unconditional prediction uses negative soft tokens during CFG. In Figure 8, it shows that injecting negative tokens into the unconditional CFG prediction produces images that may appear visually clean and high-quality, but overly suppress details and background elements which are not mentioned in the captions, resulting in overly simple images with reduced variation and consequently lower human-preference scores.

| # text (pos) / # image | ImageReward↑ | PickScore↑ | CLIP↑ | HPSv2↑ | FID↓ |
|---|---|---|---|---|---|
| 8 / 0 | 94.27 | 22.09 | 26.95 | 27.44 | **32.37** |
| 4 / 4 | 103.0 | **22.42** | **27.06** | **28.26** | 33.11 |
| 4 / 0 (ours) | **103.3** | 22.39 | 27.00 | 28.22 | 34.08 |

*Table 12.* Ablation on the number of soft tokens. # text (pos) denotes the number of $\psi^+$, and # image denotes the number of learnable soft image tokens.

| Batch size (pos:neg) | ImageReward↑ | PickScore↑ | CLIP↑ | HPSv2↑ | FID↓ |
|---|---|---|---|---|---|
| 4 (1:1) | 29.67 | 21.52 | 27.13 | 25.31 | **24.76** |
| 16 (1:3) | **34.50** | 21.59 | **27.23** | 25.66 | 25.94 |
| 64 (1:7) | 32.32 | **21.59** | 27.11 | **25.83** | 26.64 |

*Table 13.* Ablation study on the effect of batch size (SD1.5). pos:neg denotes the number of in-batch negative prompts for each positive text-image pair to calculate the guidance.

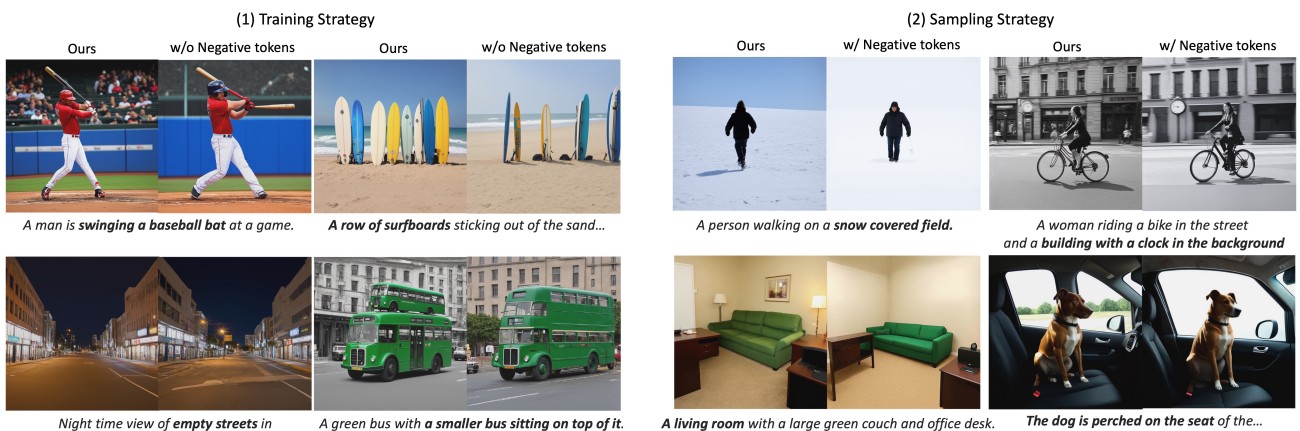

*Figure 8.* Qualitative results of the ablation study. (1) Effect of negative soft tokens optimization. (2) Effect of applying negative tokens to the unconditional prediction of CFG during sampling.

## I. Additional Qualitative Results with DPO methods

In this section, we present qualitative comparisons of DPO methods with and without soft-token integration. As shown in Figure 9, incorporating soft tokens helps the model generate images that follow the text constraints more faithfully.

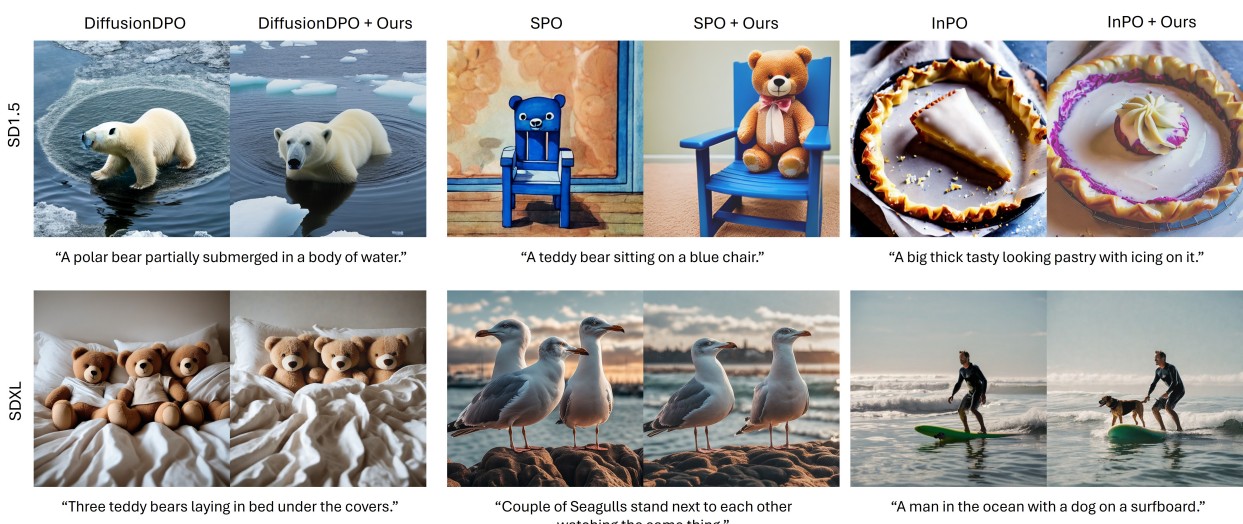

*Figure 9.* Qualitative evaluation of complementarity with other Diffusion-RL methods on COCO-val5K dataset(Lin et al., 2014).

## J. Trade-off between Text Alignment metrics and FID

In reward or preference-optimization methods for diffusion models, improving text alignment often comes with reduced diversity or coverage, which can negatively affect FID. Our results follow this general trend, but the degradation is modest relative to the base model while alignment metrics improve consistently. As shown in the main quantitative results, our method improves ImageReward, CLIP, and HPSv2 across backbones, with only a moderate FID increase compared to the base model.

Importantly, compared with SoftREPA, our method achieves a better alignment–fidelity trade-off ( Figure 5). Across SD1.5, SDXL, and SD3, AGSM obtains lower FID than SoftREPA while maintaining strong alignment performance, yielding a better ImageReward–FID Pareto front. Moreover, when combined with diffusion-RL baselines, our soft-token integration consistently improves both FID and alignment ( Table 3), suggesting that the proposed representation-level alignment can complement preference optimization without simply sacrificing image quality.

