# OpenReview forum: "Alignment-Guided Score Matching for Text-to-Image Alignment in Diffusion Models"
_ICML.cc/2026/Conference — ICML 2026 spotlight_

### Official Review · Reviewer_PF13 · 2026-03-10

**Soundness:** 2
**Presentation:** 3
**Significance:** 2
**Originality:** 3
**Overall Recommendation:** 4
**Confidence:** 2

**Summary:**

The paper proposes Alignment-Guided Score Matching (AGSM), a reward-free post-training method to improve text–image alignment in diffusion and flow-matching T2I models by optimizing a small set of soft tokens while freezing the backbone. Instead of a contrastive InfoNCE loss (as in SoftREPA), AGSM formulates alignment as a Plackett–Luce (PL) preference over in-batch text candidates, derives an explicit score-level guidance term from the model’s own conditional likelihood, and trains separate soft tokens for positive and negative pairs. Empirically, the method improves semantic faithfulness (notably counting) versus SoftREPA across SD1.5/SDXL/SD3 and shows complementarity with RL-style preference optimizers.

**Compliance With Llm Reviewing Policy:**

Affirmed.

**Key Questions For Authors:**

- Can you provide a more rigorous theoretical analysis of the approximation error introduced by deriving the reward from the $t+1$ reverse transition and substituting $x_t$ with its posterior mean? Under what specific conditions is Eq. 16 a reliable approximation of the ideal guidance?

- Did you evaluate SoftREPA using its optimal configuration (e.g., a larger number of negatives)? If the baseline was restricted to match AGSM's batch size, please provide a truly fair comparison where both models operate under their respective optimal conditions.

**Limitations:**

The authors must explicitly discuss the fidelity trade-off (e.g., the degraded FID on SD3) as a core limitation of the method. Sacrificing overall image quality for text alignment is a significant compromise that users need to be aware of. Additionally, the authors should acknowledge that relying on the model's internal reverse-transition likelihood as a "reward-free" proxy means the method cannot correct fundamental misalignments that require external human-preference signals (e.g., complex abstract concepts or subjective aesthetics).

**Strengths And Weaknesses:**

Strengths:
- Introduces an elegant score-level preference formulation for soft-token post-training. By replacing adversarial contrastive pushing with a bounded guidance term derived from PL modeling, the approach effectively mitigates known instability issues.
- The dual soft-token design ($\psi^+$ and $\psi^-$) is a simple but insightful architectural change that explicitly handles positive and negative pairs, preventing the over-emphasis on negatives seen in prior methods.
- Demonstrates solid experimental breadth across major backbones (SD1.5, SDXL, SD3) and tasks (generation and editing). The substantial improvement on GenEval counting (from 0.29 to 0.64) successfully addresses a major failure mode of SoftREPA.

Weaknesses:
- The theoretical derivations rely on strong, under-analyzed approximations. Linking the gradient of the PL term ($\nabla_{x_t}$) to the denoising error at $t+1$ via EMA involves time-shifting and posterior mean substitutions that lack rigorous error bounding. The current stability analysis relies on a very loose norm bound.
- The baseline comparisons, particularly against SoftREPA, appear poorly controlled. AGSM uses a batch size of 4 (3 negatives), while SoftREPA typically requires a larger pool of negatives to function optimally. This raises concerns that the baseline was handicapped.
- The core contributions (objective vs. architecture) are confounded. The dual-token design inherently increases the parameterization/function class compared to single-token methods.
- The method introduces noticeable trade-offs in image fidelity. For instance, on COCO-val for SD3, FID degrades compared to the base model (31.59 vs 34.08). The paper fails to deeply investigate why and when this fidelity loss occurs.

---

> ### Author Rebuttal · Authors · 2026-03-31
>
> We thank Reviewer `PF13` for the valuable feedback and clarify each of the raised points below.
>
>
> > ### **W1, Q1. Approximation errors bound on Eq.(16).**
>
> (1) Regarding time shifting, there was a typo in reverse transition on Eq.(20), so the correct form is $\log p_\theta(x_{t-1}|x_t, c)$.
> This leads Eq.(16) to
>
>  $\min_{\psi_z} E[|| \epsilon_{t,\theta}^{\psi_z}- ( \epsilon_t + \gamma_z \tilde{A_t} ( \epsilon_{t,\theta}^{ema,z} - \sum_i w_i \epsilon_{t,\theta}^{ema,i} ) )||^2 ],$
>
> without the previous time-shift ambiguity.
>
> (2) Regarding the posterior substitution, we clarify that the revised derivation does not rely on posterior mean substitution as an approximation.
> Specifically, we explicitly define the reward as (sorry for missing the expectation $E[\cdot]$ in the original submission)
>
> $r(x_t, c) = \lambda E_{q(x_{t-1}|x_t,x_0)} [ \log p_\theta(x_{t-1}|x_t, c) ]$
>
> which leads to
>
> $r(x_t, c) = E_q [ || x_{t-1} - \mu_\theta(x_t, t, c)||_2^2 ] =  || \tilde{\mu}_t - \mu_θ ||_2^2+d\tilde\beta_t.$
>
> The variance term $d\tilde\beta_t$ is independent of $\theta$ and can therefore be omitted. As a result, the reward is proportional to
>
> $r(x_t,c)\propto -\||\epsilon_\theta(x_t,t,c)-\epsilon_t\||_2^2.$
>
> (3) Regarding EMA, it is used only as a practical stabilizer for reward evaluation, to reduce high-variance fluctuations. Once training converges, any discrepancy due to EMA vanishes.
>
> We will revise the paper to make these points explicit: the reward derivation is now written in a fully explicit form, the timestep misalignment in the previous draft has been corrected.
>
> ---
>
> > ### **W2, Q2. Fair comparison against SoftREPA.**
>
> We would like to clarify that we did not re-train or restrict SoftREPA under a suboptimal configuration.  We evaluate using the official checkpoints, trained with larger negative pools: 7 negatives per positive (batch 64) for SD1.5/SDXL, and 3 negatives (batch 16) for SD3. In contrast, our method uses a unified setting of 3 negatives per positive (batch 16) across all models. Thus, the comparison does not advantage AGSM. Rather, AGSM remains competitive with SoftREPA while using a smaller batch size for SD1.5/SDXL. Our goal is to show that AGSM performs strongly in general settings without relying on large negative pools, while remaining competitive with SoftREPA under its original configurations. We will clarify this in the paper.
>
> ---
>
> > ### **W3. Clarification on Contribution.**
>
> We would like to clarify that our core contribution relies on: (1) reward-free PL formulation, (2) bounded negative guidance and (3) dual token design.
>
> To better articulate our contributions, we add controlled ablations based on Eq.(18), where the first term trains positive tokens and the second trains negative tokens (Table R10). (a) uses only the positive objective with the same total number of tokens, (b) uses both objectives but without separating tokens, and (c) is our full method with dual objectives and dual tokens. The results show that the gain is not solely from adding the negative objective (a→b), but also from the structured dual-token design (b→c).
>
> **Table R10. Architecture and Objective Contribution Ablation**
> | # pos / # neg | ImageReward | PickScore| CLIP | HPSv2| FID |
> |:---|:------------------|:-------:|:-------:|:-------:|:------:|
> | (a) 8 / 0 | 94.79 | 22.26 | 26.93 | 27.81 | 34.46 |
> | (b) 8 (shared) | 47.33 | 21.69 | 25.68 | 25.82 | **31.20** |
> | (c) 4 / 4 (ours) | **103.3** | **22.39** | **27.00** | **28.22** | 34.08 |
>
> ---
>
> > ### **W4. Trade-off between Text Alignment and FID.**
>
> Please refer to the response to reviewer `1nTa`’s W1. Our method achieves a better alignment–fidelity trade-off: while FID increases only slightly relative to the base model, alignment metrics consistently improve. Compared to SoftREPA, we obtain lower FID and a better Pareto front.

---

> > ### Author Rebuttal · Reviewer_PF13 · 2026-04-02
> >
> > Thank the authors for their response. My concerns have been addressed. I will keep my score, good luck.

---

> > > ### Author Response · Authors · 2026-04-02
> > >
> > > We're pleased to hear that your earlier concerns have been successfully resolved. We greatly appreciate your feedback and the time dedicated to reviewing our work.

---

### Official Review · Reviewer_yiH5 · 2026-03-11

**Soundness:** 3
**Presentation:** 2
**Significance:** 3
**Originality:** 3
**Overall Recommendation:** 5
**Confidence:** 3

**Summary:**

This paper proposes Alignment-Guided Score Matching, a reward-free post-training method for improving text-to-image alignment in diffusion models by optimizing a small set of soft text tokens. The main idea is to replace SoftREPA’s contrastive objective with an alignment-guided score objective, with the goal of avoiding the simultaneous optimization of negative pairs and providing a more balanced training signal. Experiments on Stable Diffusion variants show consistent improvements over SoftREPA on alignment-related metrics, including the counting benchmark in GenEval.

**Compliance With Llm Reviewing Policy:**

Affirmed.

**Final Justification:**

I recommend acceptance of this work. The proposed methodology is novel and addresses a timely problem in this line of research. Although I initially had several concerns, the authors fully addressed them during the rebuttal process. Furthermore, the training analysis provided in the rebuttal to Reviewer iy3t further strengthens the impact of the framework. Therefore, I have decided to raise my score.

**Key Questions For Authors:**

See Weaknesses.

**Limitations:**

yes

**Strengths And Weaknesses:**

### **Strengths**
1. The paper is well motivated. It identifies a plausible limitation of the contrastive objective used in prior work and introduces an alternative alignment-guided score objective that is conceptually more targeted.

2. The method is mostly clearly described, with a concrete algorithm and a helpful schematic that connects the proposed objective to the modified score-matching formulation.

3. The experimental section is reasonably extensive and suggests that the method is stable in practice. The ability to combine the approach with other preference-optimization methods is also practically valuable.

### **Weaknesses**
Overall, I find the proposed idea interesting and promising. However, I think the paper would benefit from stronger presentation and more complete empirical analysis to really make the contribution standout. I have several concerns as below:

1. It is not fully clear to me why the negative region is explicitly modeled with a negative-sign weight in Eq.10.

2. The reward is formulated based on the DDPM posterior distribution. This raises the question of whether the formulation remains appropriate under non-consecutive or jumping-step inference, such as DDIM. I encourage the authors to discuss how the method extends to more general sampling and training settings.

3. The stability discussion in Lines 256–273 does not seem sufficient to explain the superiority of the new objective compared to SoftREPA. My understanding is that the candidate weight $w_i$ plays a central role in the alignment of the positive prompt. When the noisy latent state and the prompt are well aligned, this weight may approach 1, in which case the denoising error could become arbitrarily small. However, the paper does not provide empirical evidence (e.g., $w_i$ during training) to support this interpretation.

4. There are several places where the notation is difficult to follow. For example, the role of $\gamma_z$, which appears to involve an indicator function, is not clearly explained in terms of when and how it is applied; the distinction between $z$ vs $\mathbf{z}$ should be defined more carefully; the typo errors in Eq. 3 (e.g., $\mathbf{c}^l$, $\mathbf{z}_t^l$, etc.)

5. The paper is missing sensitivity analyses for key hyperparameters $\gamma^+$,$\gamma^-$, and $\lambda$.

6. The sensitive analysis with the batch size, which corresponds to the negative prompt pool size is also not reported.

---

> ### Author Rebuttal · Authors · 2026-03-31
>
> We appreciate Reviewer `yiH5`’s constructive comments and respond to each point below.
>
> ---
> > ### **W1. Clarification about the negative sign weight.**
>
>  Eq.(10) yields a unified objective with attractive guidance for matched pairs and repulsive guidance for mismatched pairs. This is consistent with the intuition behind negative prompting in diffusion models, where the same conditional signal is used with the opposite direction to suppress undesired semantics [1,2,3,4].
>
> More specifically, since $p(z \mid x_t, c)$ measures compatibility between $x_t$ and $c$, the positive branch increases the density of high-alignment regions, while the negative branch suppresses them.
>
> ---
> > ### **W2. Extension to general sampling settings.**
>
> While the reward in the paper is derived from the reverse transition $q(x_{t-1}\mid x_t,x_0)$ (please refer to reviewer `PF13`'s W1), which is Gaussian in DDPM and in stochastic DDIM ($\sigma_t>0$), this reward depends only on the denoising objective, not on the posterior. Therefore, after post-training, the learned score (or vector field) can be applied to various samplers. More generally, our alignment mechanism operates on the learned score (or vector field), while the sampler (e.g., DDPM, DDIM, or other variants) only determines how this field is traversed at inference time.
>
> Additionally, in our implementation, we found that uniform weighting of $\tilde A(t)$ in Eq.(16) across timesteps works well in practice. Under this simplification, the reward reduces to a form proportional to the negative denoising (score-matching) loss, $r_t(x_t,c)\propto -||\epsilon_\theta(x_t,t,c)-\epsilon_t||^2$, which again confirms the independence of any posterior parameterization.
>
> We will revise the paper to clarify this distinction between the derivation and the implementation.
>
> ---
> > ### **W3. Empirical evidence on training stability.**
>
> In our formulation, the reviewer is kindly reminded that the alignment correction enters through $\Delta=\epsilon(x_t,c)-\sum_i w_i\epsilon(x_t,c_i)$ in Eq.(16). When the matched candidate becomes dominant, $\Delta$ becomes small, so the target approaches the standard denoising target. For negative pairs, $\Delta$ remains a bounded and normalized guidance term, rather than encouraging increasingly large negative denoising errors as may happen in SoftREPA's contrastive objective.
>
> To support this intuition empirically, we additionally analyze training stability; please see our response to reviewer `iy3t`'s W2.
>
> ---
> > ### **W4. Clarification on notation.**
>
> Thank you for pointing out the typo. Our intended meaning is that $z$ is a binary event variable: $z=1$ is aligned/preferred, and $z=0$ otherwise.
>
> To remove any ambiguity in the revision, we will:
>
> - write probabilities explicitly as $p(z=1\mid\cdot)$ or $p(z=0\mid\cdot)$, instead of $p(z\mid\cdot)$;
> - revise the text around Eq. (3) and Section 3.1 so that $z$ is defined only as a binary event variable, not as a probability or alignment score;
> - clarify that $\gamma_z$ is shorthand for $\gamma^{+}\mathbf{1} \(z=1\)-\gamma^{-}\mathbf{1}\(z=0\)$, and state explicitly where this indicator applies.
>
> ---
> > ### **W5. Hyperparameter sensitivity.**
>
> Regarding the sensitivity of the negative guidance scale, please refer to reviewer `1nTa`’s Q2.
>
> Regarding $\lambda$, it is introduced mainly for consistency with the preliminary reward formulation, and in practice, we fix it to $\lambda=1$.
>
> ---
> > ### **W6. Ablation study on batch size.**
>
> In response to the reviewer’s request, we conduct an ablation study on batch size using COCO dataset (Table R9). We observe that increasing the batch size from 4 to 16 leads to clear improvements across alignment-related metrics. However, further increasing it from 16 to 64 yields only marginal additional gains. These results suggest that a larger negative pool is helpful up to a certain point, but the method is not overly dependent on very large batch sizes.
>
> **Table R9. Ablation on the batch size (SD1.5)**
> | Batch size (pos:neg) | ImageReward | PickScore| CLIP | HPSv2| FID |
> |:-----------|:-------------:|:--------:|:------:|:-----:|:-----:|
> | 4 (1:1) | 29.67 | 21.52 | 27.13 | 25.31 | **24.76** |
> | 16 (1:3) | **34.50** | 21.59 | **27.23** | 25.66 | 25.94 |
> | 64 (1:7) | 32.32 | **21.59** | 27.11 | **25.83** | 26.64 |
>
> ---
> **Reference**
>
> [1] Armandpour, Mohammadreza, et al. "Re-imagine the negative prompt algorithm: Transform 2d diffusion into 3d, alleviate janus problem and beyond." arXiv preprint arXiv:2304.04968 (2023).
>
> [2] Gandikota, Rohit, et al. "Erasing concepts from diffusion models." Proceedings of the IEEE/CVF international conference on computer vision. 2023.
>
> [3] Koulischer, Felix, et al. "Dynamic negative guidance of diffusion models." arXiv preprint arXiv:2410.14398 (2024).
>
> [4] Liu, Nan, et al. "Compositional visual generation with composable diffusion models." European conference on computer vision. Cham: Springer Nature Switzerland, 2022.

---

> > ### Author Rebuttal · Reviewer_yiH5 · 2026-04-02
> >
> > Thank the authors for their response. My concerns have been addressed. Additionally, the analysis of the training dynamics sounds interesting. The authors should include it in the final version. I will raise my score.

---

> > > ### Author Response · Authors · 2026-04-02
> > >
> > > Thank you very much for raising your score. We're glad that our rebuttal successfully addressed your earlier concerns, and we will ensure that the analysis of training dynamics is included in the final version.

---

### Official Review · Reviewer_1nTa · 2026-03-12

**Soundness:** 3
**Presentation:** 3
**Significance:** 3
**Originality:** 3
**Overall Recommendation:** 5
**Confidence:** 3

**Summary:**

To address the problem of SoftREPA excessively penalizing negative pairs, this paper proposes an alignment-guided score matching method that embeds contrastive alignment guidance into the score-matching objective of diffusion models. It constructs a reward-free preference learning framework based on the Plackett-Luce model, thus preventing the model from imposing unbounded penalties on negative pairs. The paper assigns explicit score-level guidance to positive and negative pairs via independent positive and negative soft tokens, and also designs a dual-token training scheme to ensure the stability of optimization. The proposed method is validated on mainstream diffusion models and yields a more than 35% improvement in counting accuracy on the GenEval benchmark. In addition, it is complementary to existing reinforcement learning-based post-training methods for diffusion models, and features light weight and model agnosticism. It effectively mitigates the off-manifold deviation issue of previous methods and enhances the semantic consistency of text-to-image generation and editing.

**Compliance With Llm Reviewing Policy:**

Affirmed.

**Final Justification:**

The authors have addressed my concerns in the rebuttal. Based on this, I update the score to 5.

**Key Questions For Authors:**

1. All experimental configurations in the paper are consistent with those of SoftREPA, such as the use of 4 soft text tokens. Have the authors explored other values for this hyperparameter?

2. The paper sets different positive and negative guidance scales for different models. For non-COCO datasets, do there exist universal settings for these scales that require no significant hyperparameter tuning?

**Limitations:**

yes

**Strengths And Weaknesses:**

Strengths:
1. The research motivation of the paper is clear, and targeted improvements are proposed.
2. The method presented is simple and straightforward, and theoretically addresses the flaws of SoftREPA.
3. The experimental section validates the method on multiple diffusion model backbones and against various post-training approaches.

Weaknesses:
1. In the COCO-val experiments, the FID metric saw a slight increase across all three models (notably, the FID of SD3 rose from 31.59 to 34.08). The authors did not conduct a quantitative analysis or provide a causal explanation for the trade-off between the improved alignment accuracy and the slight degradation in image quality.
2. The model was trained solely on the COCO dataset, with no exploration of more practical scenarios such as long text descriptions.

---

> ### Author Rebuttal · Authors · 2026-03-31
>
> We thank Reviewer `1nTa` for the thoughtful feedback and provide detailed responses to each concern below.
>
> ---
> > ### **W1. Trade-off between Text Alignment metrics and FID.**
>
> In reward or preference-optimization methods for diffusion models, improving alignment often comes at the cost of reduced diversity or coverage, which can negatively affect FID, as noted in prior work [1]. Similarly, our results follow this general trend, but the increase in FID is modest relative to the base model, while alignment improves consistently (main Table 1). Compared to SoftREPA, we achieve lower FID across all backbones and a better ImageReward–FID Pareto front (main Figure 4). Moreover, when combined with diffusion-RL baselines (main Table 3), our method consistently improves both FID and alignment, supporting better overall quality.
>
> ---
> > ### **W2. Evaluation on long text prompts.**
>
> In response to the reviewer’s suggestion, we evaluate on UniGenBench++ [2] for long-text T2I generation with 600 prompts, comparing against SD3 and SoftREPA. As shown in Table R7, our method achieves better human preference and text alignment, demonstrating practical effectiveness.
>
> **Table R7. Comparison on UniGenBench++**
> | Model | ImageReward | CLIP | PickScore | HPSv2 |
> |:---|:---:|:---:|:---:|:---:|
> | SD3 Base | 82.33 | 29.50 | 21.32 | 28.87 |
> | SoftREPA | 90.63 | 28.93 | 21.18 | 28.61 |
> | Ours | **98.01** | **29.56** | **21.48** | **29.66** |
>
>
> ---
> > ### **Q1. Ablation study on the number of soft text tokens.**
>
> For fair comparison with SoftREPA, we use 4 soft text tokens. Performance of SoftREPA is known to be stable around 4 tokens, and starts to degrade with 8 (Table 10 of SoftREPA). We confirm the similar behavior in our setting (Table R8). Interestingly, unlike SoftREPA which fails with image tokens, our method successfully optimizes both text and image tokens, demonstrating improved flexibility (Table R8).
>
> **Table R8. Ablation on the number of soft tokens**
> | # text/ # image | ImageReward | PickScore| CLIP | HPSv2| FID |
>  |:---------|:-------------:|:--------:|:--------:|:------:|:-----:|
> | 8 / 0 | 94.27 | 22.09 | 26.95 | 27.44 | **32.37** |
> | 4 / 4 | 103.0 | **22.42** | **27.06** | **28.26** | 33.11 |
> | 4 / 0 (ours) |**103.3** | 22.39 | 27.00 | 28.22 | 34.08 |
>
> ---
> > ### **Q2. Clarification on hyperparameter scales.**
>
> Once set during training, the negative guidance scale generalizes across datasets (e.g., GenEval, DIV2K) and tasks (e.g., generation, editing) without retraining. While the scale is model-dependent, a simple rule works well in practice: we employ a larger value (\~1.0) for models with stronger CFG (e.g., SD1.5/SDXL) and a smaller value (\~0.1) for weaker CFG models (e.g., SD3), avoiding dataset-specific tuning.
>
> Additionally we conduct a sensitivity analysis on SD3 with $\gamma^- \in \{0.05, 0.1, 0.5, 1.0\}$ (Table R6).
> The results show that performance remains stable across this range, with the smaller value (~0.1) achieving the best alignment performance.
>
> **Table R6. Comparison on negative scale guidance**
> | $\gamma^-$ | ImageReward | PickScore| CLIP | HPSv2| FID |
>  |:---------|:-------------:|:--------:|:------:|:-----:|:-----:|
> | 1 | 96.71 | 22.36 | 26.78 | 28.22 | 35.21 |
> | 0.5 | 98.44 | 22.31 | **27.02** | 27.96 | 33.88 |
> | 0.1 (ours) |**103.3** | 22.39 | 27.00 | **28.22** | 34.08 |
> | 0.05  | 100.1  |  **22.46** | 26.95  | 28.11  | **33.04**  |
> | 0 | 94.79 | 22.26 | 26.93 | 27.81 | 34.46 |
>
> ---
> **Reference**
>
> [1] Jena, Rohit, et al. "Elucidating optimal reward-diversity tradeoffs in text-to-image diffusion models." 2025 IEEE/CVF Winter Conference on Applications of Computer Vision (WACV). IEEE, 2025.
>
> [2] Wang, Y., Li, Z., Zang, Y., Zhou, Y., Bu, J., Wang, C., ... & Wang, J. (2025). Pref-grpo: Pairwise preference reward-based grpo for stable text-to-image reinforcement learning. arXiv preprint arXiv:2508.20751.

---

> > ### Author Rebuttal · Reviewer_1nTa · 2026-04-06
> >
> > The authors have addressed my concerns in the rebuttal. Based on this, I will update my score.

---

> > > ### Author Response · Authors · 2026-04-07
> > >
> > > Thank you for raising your score. We're glad to hear that your previous concerns have been addressed. We sincerely appreciate your feedback and the time you invested to review our work.

---

### Official Review · Reviewer_iy3t · 2026-03-13

**Soundness:** 3
**Presentation:** 3
**Significance:** 2
**Originality:** 2
**Overall Recommendation:** 4
**Confidence:** 3

**Summary:**

The key idea of this paper is to replace SoftREPA-style contrastive soft-token tuning with an alignment-guided score-matching objective, where alignment is modeled via a Plackett-Luce preference formulation.

**Compliance With Llm Reviewing Policy:**

Affirmed.

**Final Justification:**

My concerns have been addressed. I have raised my score to 4.

**Key Questions For Authors:**

How sensitive is the method to the negative-guidance scale and the negative sampling strategy? The paper uses (1,1) for SD1.5/SDXL and (1,0.1) for SD3, which suggests some backbone-dependent tuning.

**Limitations:**

See my weakness

**Strengths And Weaknesses:**

Strengths:
The paper is well motivated and targets a concrete failure mode of prior contrastive soft-token methods.
The experimental evaluation is comprehensive, covering generation, editing, and combinations with DPO-style post-training methods.

Weaknesses:

1. My major concern is that the empirical gains are somewhat not convincing. The paper works well on counting, but not better than SoftREPA on all subtasks, especially in GenEval. In Table 1, SoftREPA is still better on some GenEval subtasks like Two, Position, and Color Attribution. Likewise, on COCO-val, SoftREPA often has a higher ImageReward while the paper has better FID.

2. The paper provides a boundedness argument showing that its PL form is bounded and more stable, in contrast to the implicit negative pushing induced by the SoftREPA contrastive term. However, it lacks a rigorous characterization of training dynamics or convergence. In particular, the paper does not show when or to what extent SoftREPA is unstable in practice.

3. Why is the Plackett-Luce formulation necessary? Discussion with some other simpler alternatives, such as pairwise Bradley-Terry-style ranking or a simpler batch softmax formulation, is needed to understand the contribution.

The paper looks interesting, and I'm willing to raise my score if the authors can address my concerns.

---

> ### Author Rebuttal · Authors · 2026-03-31
>
> We sincerely thank Reviewer `iy3t` for the constructive comments. Below, we address each of the raised questions and concerns in detail:
>
> ---
> > ### **W1. Not surpassing SoftREPA on some subtasks / metrics.**
>
> While the contrastive formulation of SoftREPA can improve certain aspects of alignment, we would like to kindly remind the reviewer that its performance significantly degraded in some other tasks such as counting and repetition. Our primary goal is therefore to address this key failure mode and improve the overall performance.
>
> As shown in main Table 1, our method significantly improves counting on GenEval (0.29 -> 0.64), where SoftREPA degrades the pretrained model (0.56 -> 0.29). Main Figure 4 further shows that our method achieves a better ImageReward-FID trade-off. We observe a similar pattern in image editing, and this is further supported by UnifiedReward [1] (Table R1) and our user study (Table R2), both of which favor our method over SoftREPA in overall quality.
>
>
> **Table R1. Quantitative evaluation of T2I generation on SD3 on the COCO-val 5K**
> | Model | Unified Reward | CLIP | FID |
> |:---|:---:|:---:|:---:|
> | SD3 Base | 4.1130 | 26.30 | **31.59** |
> | SoftREPA | 4.0922 | 26.91 | 36.21 |
> | Ours | **4.1370** | **27.00** | 34.08 |
>
>
> **Table R2. User study results on SD3**
> | Model | Visual Naturalness | Text Alignment | Overall Preference |
> |:---|:---:|:---:|:---:|
> | SoftREPA | 22.66 | 15.33 | 19.22 |
> | Ours | **63.28** | **52.11** | **65.49** |
> | Draw | 14.06 | 32.57 | 15.29 |
>
> ---
> > ### **W2. Training stability compared to SoftREPA.**
>
> To address the reviewer’s concern, we track the training loss value and validation ImageReward metric of two methods during training.
>
> As shown in Table R3, SoftREPA peaks early (10k–15k iterations) and then degrades, while our method improves more steadily without late-stage deterioration. Table R4 further shows that SoftREPA’s training loss continues to decrease even as ImageReward drops, indicating over-optimization and the need for heuristic early stopping, whereas our method maintains stable improvements.
>
> These trends confirm the advantage of our method: our PL-based objective introduces a bounded, normalized correction (Eq. (19)), while SoftREPA’s contrastive loss can amplify negative signals, leading to less stable training.
>
>
> **Table R3. Training Dynamics Comparison between SoftREPA and Ours**
> | Iteration | Softrepa ImageReward | Ours ImageReward |
> |:---|:---:|:---:|
> | 5k | 89.40 | 87.78 |
> | 10k | **95.87** | 94.76 |
> | 15k | 95.38 | 93.54 |
> | 20k | 94.65 | 93.87 |
> | 30k | 88.91 | 94.29 |
> | 40k | 75.31 | **98.08** |
> | 50k | 84.46 | 93.48 |
> | 60k | 76.09 | 94.35 |
> | 70k | - | 98.03 |
> | 80k | - | 94.84 |
>
>
> **Table R4. Loss value and ImageReward of SoftREPA**
> | Iteration | SoftREPA Loss | SoftREPA ImageReward |
> |:-----------|:-------------:|:------------------:|
> | 5k | 1.3780 | 89.40 |
> | 10k | 1.3768 | **95.87** |
> | 15k | 1.3754 | 95.38 |
> | 20k | 1.3745 | 94.65 |
> | 30k | 1.3690 | 88.91 |
> | 40k | 1.3562 | 75.31 |
> | 50k | 1.3525 | 84.46 |
> | 60k | 1.3525 | 76.09 |
>
> ---
> > ### **W3. Missing comparison on simpler alternatives other than the PL formulation.**
>
> The reviewer is kindly reminded that the Plackett–Luce (PL) [2] formulation is the natural multi-candidate extension of Bradley–Terry (BT).  In fact, a simpler batch softmax formulation is closely related to PL. Our setting involves multiple in-batch negatives, whereas BT compares a single positive-negative pair for pairwise pushing.
>
> We compare pairwise BT with our PL-based formulation using three negative prompts in the batch (Table R5). Our method yields substantially better alignment performance across ImageReward, PickScore, CLIP, and HPSv2, while BT attains slightly better FID, demonstrating PL’s advantage in multi-candidate alignment and alignment quality.
>
>
> **Table R5. Comparison between BT and PL model**
> |           | ImageReward | PickScore| CLIP | HPSv2| FID |
>  |:---------|:-------------:|:--------:|:------:|:-----:|:-----:|
> | BT | 29.67 | 21.52 | 27.13 | 25.31 | **24.76** |
> | PL(ours) | **34.50** | **21.59** | **27.23** | **25.66** | 25.94 |
>
> ---
> > ### **Q1. Sensitivity of the negative guidance scale / negative sampling strategy.**
>
> Regarding the sensitivity of the negative guidance scale, please refer to reviewer `1nTa`'s Q2.
>
> Additionally, we want to clarify that our method does not introduce additional sampling strategy for negative pairs, adopting standard in-batch negative construction.
>
> ---
> **Reference**
>
> [1] https://github.com/CodeGoat24/UnifiedReward/blob/main/inference_qwen/image_generation/qwen_point_score_image_generation.py
>
> [2] R. L. Plackett, The Analysis of Permutations, Journal of the Royal Statistical Society Series C: Applied Statistics, Volume 24, Issue 2, June 1975, Pages 193–202, https://doi.org/10.2307/2346567

---

> > ### Author Rebuttal · Reviewer_iy3t · 2026-04-01
> >
> > My concerns are solved. So, I will raise my score.

---

> > > ### Author Response · Authors · 2026-04-01
> > >
> > > Thank you very much for increasing your score. We’re pleased to hear that your original concerns have been properly addressed.

---

### Decision · Program_Chairs · 2026-04-30

**Decision:**

Accept (spotlight)

**Comment:**

AC agrees with the Rs that this paper proposes Alignment-Guided Score Matching, a reward-free post-training method that replaces SoftREPA's contrastive objective with a Plackett-Luce preference formulation to improve text-to-image alignment in diffusion models. Reviewers appreciated the clear motivation targeting SoftREPA's concrete failure modes (over-counting, repetition), the elegant dual soft-token design, the substantial improvement on GenEval counting accuracy and comprehensive evaluation across SD1.5, SDXL, and SD3 backbones demonstrating complementarity with RL-based methods. Initial concerns about empirical gains not being uniformly convincing across all subtasks, training stability compared to SoftREPA, necessity of the Plackett-Luce formulation over simpler alternatives, theoretical approximation rigor, baseline comparison fairness, and hyperparameter sensitivity were thoroughly addressed in rebuttal through training dynamics analysis showing SoftREPA's early peaking and degradation, Bradley-Terry vs Plackett-Luce ablations, clarification of reward derivation with corrected notation, confirmation that official SoftREPA checkpoints with larger negative pools were used, and sensitivity analyses on negative guidance scale and batch size. All four reviewers confirmed their concerns were fully resolved, with two raising scores to 5 (accept) and two maintaining scores of 4 (weak accept). While the paper has some limitations including slight FID degradation as a trade-off for improved alignment and model-dependent hyperparameter tuning for negative guidance scale, the work makes a solid contribution. AC recommends accept.